# The hippocampus encodes delay and value information during delay-discounting decision making

Akira Masuda[1,2]*, Chie Sano[1], Qi Zhang[1,3], Hiromichi Goto[1], Thomas J McHugh[4], Shigeyoshi Fujisawa[5], Shigeyoshi Itohara[1]*

[1]Laboratory for Behavioral Genetics, Center for Brain Science, RIKEN, Wako, Japan; [2]Organization for Research Initiatives and Development, Doshisha University, Kyotanabe, Japan; [3]Faculty of Human Science, University of Tsukuba, Tsukuba, Japan; [4]Laboratory for Circuit and Behavioral Physiology, Center for Brain Science, RIKEN, Wako, Japan; [5]Laboratory for Systems Neurophysiology, Center for Brain Science, RIKEN, Wako, Japan

**Abstract** The hippocampus, a region critical for memory and spatial navigation, has been implicated in delay discounting, the decline in subjective reward value when a delay is imposed. However, how delay information is encoded in the hippocampus is poorly understood. Here, we recorded from CA1 of mice performing a delay-discounting decision-making task, where delay lengths, delay positions, and reward amounts were changed across sessions, and identified subpopulations of CA1 neurons that increased or decreased their firing rate during long delays. The activity of both delay-active and -suppressed cells reflected delay length, delay position, and reward amount; but manipulating reward amount differentially impacted the two populations, suggesting distinct roles in the valuation process. Further, genetic deletion of the N-methyl-D-aspartate (NMDA) receptor in hippocampal pyramidal cells impaired delay-discount behavior and diminished delay-dependent activity in CA1. Our results suggest that distinct subclasses of hippocampal neurons concertedly support delay-discounting decisions in a manner that is dependent on NMDA receptor function.

*For correspondence:
amasuda@mail.doshisha.ac.jp (AM);
shigeyoshi.itohara@riken.jp (SI)

**Competing interests:** The authors declare that no competing interests exist.

**Reviewing editor:** Matthijs van der Meer,

## Introduction

Animals faced with multiple options optimize their decisions through a complex cost-benefit valuation. The introduction of a time delay decreases preference for the delayed option (delay discounting) (*Ainslie, 1992*; *Ainslie, 1975*), with the discount rate varying on an individual basis. People who are considered patient exhibit lower discount rates, whereas impatient (or impulsive) people exhibit higher discount rates. Further, higher discount rates have been shown to be related to various neuropsychological disorders (*Bickel and Marsch, 2001*; *Chesson et al., 2006*; *Epstein et al., 2008*; *Luman et al., 2010*; *Odum et al., 2000*; *Weller et al., 2008*). Although lesion studies have revealed a critical role for the hippocampus in delay discounting (*Figner et al., 2010*; *Kalivas and Volkow, 2005*; *Peters and Büchel, 2011*; *Cheung and Cardinal, 2005*; *Mariano et al., 2009*; *McHugh et al., 2008*), how this is reflected in hippocampal activity remains poorly understood.

Decades of study point to a critical role for the hippocampus in episodic memory (*Scoville and Milner, 1957*; *Squire, 1992*) and spatial navigation (*Burgess et al., 2002*; *Ekstrom et al., 2003*). Although much of the rodent hippocampal physiology literature has focused on the spatial code present in hippocampal place cell activity (*Jung and McNaughton, 1993*; *O'Keefe and Dostrovsky, 1971*; *Wilson and McNaughton, 1993*), subsequent work has demonstrated that the circuit is capable of encoding a variety of spatiotemporal features beyond the animal's current position, including

past and future trajectories (*Ambrose et al., 2016*; *Foster and Wilson, 2006*; *Johnson et al., 2007*; *Johnson and Redish, 2007*; *Pfeiffer and Foster, 2013*; *Skaggs and McNaughton, 1996*), the location of other animals or objects (*Danjo et al., 2018*; *Omer et al., 2018*), internal time (*Kraus et al., 2013*; *MacDonald et al., 2011*; *Manns et al., 2007*; *Pastalkova et al., 2008*), and various physical scales (*Aronov et al., 2017*; *Terada et al., 2017*).

When trying to understand the links between behavior and physiological data, several factors must be considered, including the variable(s) correlated with the activity, the regions or cell assemblies engaged and the mechanisms of representation on both the single-cell and population levels. To this end, studies combining imaging and recording with optogenetic manipulation and identification suggest that subsets of CA1 neurons can encode distinct features of a task (*Cembrowski et al., 2016*; *Danielson et al., 2016*). Consistent with these data, *Gauthier and Tank (2018)* recently identified a unique population of neurons that are active at reward sites, serving as 'reward cells'. Although the modulation of reward-based activity has been well-investigated in relation to spatial context (*Hölscher et al., 2003*; *Lee et al., 2012*; *Murty and Adcock, 2014*; *Ólafsdóttir et al., 2015*; *Peters and Büchel, 2010*; *Singer and Frank, 2009*) or probability discounting (*Tryon et al., 2017*), the impact of delay (length/location) and reward amount, which both alone and together constitute the core computation for delay-based decision making, has not been examined. Further, since delay and reward both modulate value, the most important parameter for decision making, neurons encoding value information should respond in similar ways to changes in delay length increment/decrement and reward amount loss/gain, as can be seen in dopaminergic neurons in the ventral tegmental area (VTA) (*Roesch et al., 2007*).

There are at least two dissociable schemes of hippocampal coding for delay length. One is on the population level, with time cells (*MacDonald et al., 2011*; *Pastalkova et al., 2008*) — which are a series of neurons that separately represent distinct temporally receptive fields that tile the delay period — forming sequences that correspond with different delay lengths. The other is rate coding, where individual neurons change their firing rate according to the delay length variations.

Here, we designed experiments to identify and characterize hippocampal neurons that are engaged during the delay of a delay-discounting task and to probe their sensitivity to changes in delay length, delay position, and reward amount. We recorded single-unit activity in CA1 of mice performing a delay-discounting version of the T-maze task (*Zhang et al., 2018*), and assessed changes in neural activity related to delay length, delay position, and reward size. We first examine the two schemes for encoding delay length: population coding and rate coding, and found that both schemes were employed by a significant fraction of CA1 neurons, including two populations that demonstrated increased or decreased delay-period activity. The activity of these distinct populations reflected delay length, delay positions, and/or reward size, however manipulation of reward size resulted in these populations' having opposite responses. Finally, we were able to identify a specific population of neurons that fit the criteria of value coding. These results suggest that distinct subpopulation of neurons in the hippocampus can have unique contributions to the valuation processes that are required for delay-based decisions.

## Results

### Behavioral profiling

We conducted a delay-based decision-making task in mice using the T-maze, in which mice chose between right or left goal arms, with each arm containing a small reward or large reward, and with or without delay, respectively (*Figure 1A*). In total, we employed five behavioral conditions, with each session consisting of about 10 trials within 30 min, with a 20 s inter-trial interval at the start zone.

To examine the impact of delay on decision making, we changed the delay length sequentially. Once mice showed a preference for the large reward arm (>80%) we increased the length of delay in a stepwise fashion (e.g., 0, 5, 10, 20, and 40 s; *Figure 1B*; *Figure 1—figure supplement 1*). With the inclusion of a delay, preference for the large reward arm decreased as a function of delay length (*Figure 1C*). The whole schedule of experiments is shown in *Figure 1—figure supplement 2*.

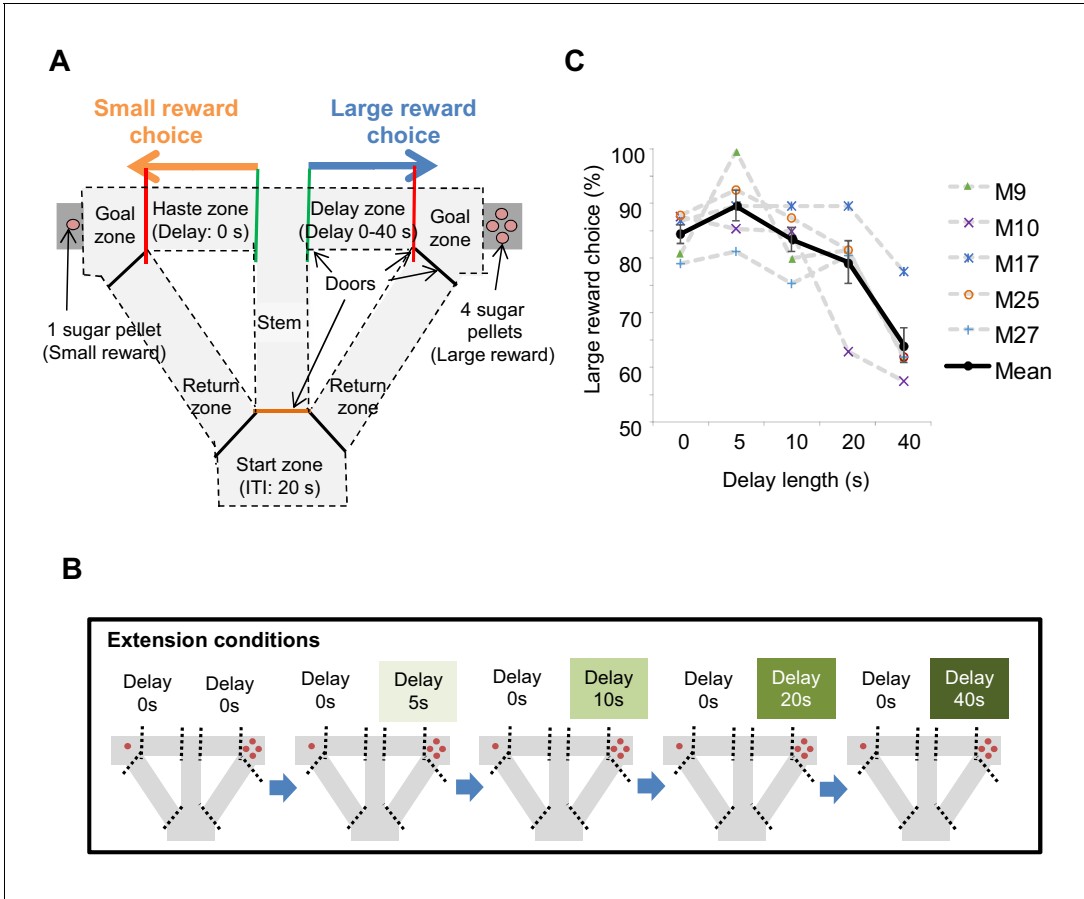

**Figure 1.** Task design of the delay-based decision making in the T-maze. (**A**) Schematic diagram for the experimental setup. Mice can choose the right or left arms assigned to obtain the small reward without delay or the large reward with delay, respectively. (**B**) Flow of the extension conditions. The delay lengths were extended sequentially. Red circles indicate the number of sugar pellets. (**C**) Percentages of large-reward choices as a function of delay length. Error bars indicate the standard error of the means (SEM).

The online version of this article includes the following source data and figure supplement(s) for figure 1:

**Source data 1.** Source Data File for *Figure 1C*.
**Figure supplement 1.** Daily timeline of the extension of delay conditions.
**Figure supplement 2.** Schedule of running experimental conditions.

## Delay-dependent neuronal activity in CA1

We recorded extracellular single units and local-field potentials (LFPs) from the CA1 region in a total of 28 mice (*Figure 2—figure supplement 1*) during delay-discount behavior, and classified cells as putative excitatory neurons or inhibitory neurons on the basis of the characteristics of their extracellular waveform (*Figure 2—figure supplement 2*; see 'Materials and methods'). We first analyzed LFP signals in the CA1 region during delay periods. Consistent with the active movement of the mice, sharp-wave/ripples (SWRs) were rarely observed ($Z=2.19$, $p=0.03$, for start and stem zones vs delay zone; $Z=2.40$, $p=0.02$, for delay zone vs goal zone; Mann-Whitney $U$ Test; *Figure 2A and B*) and the LFP was dominated by theta-range (7–11 Hz) activity (*Figure 2C and D*), suggesting that the circuit remained engaged during this phase. We then examined the activity of excitatory neurons (*Table 1*) during specific task events, the exit from the start zone (start), the entrance/exit of the delay zone (delay), and the entrance to the goal zone (goal). In CA1, a subset of neurons exhibited delay-related activity, with firing rate rising during longer delays (>20 s) (*Figure 3A*), whereas a distinct subset fired only under short delay conditions, decreasing their firing rate as the delay length increased (*Figure 3B*).

We next asked whether neurons significantly altered their firing rate during long delays compared with other phases of the task (see 'Materials and methods'). We found that across all task conditions,

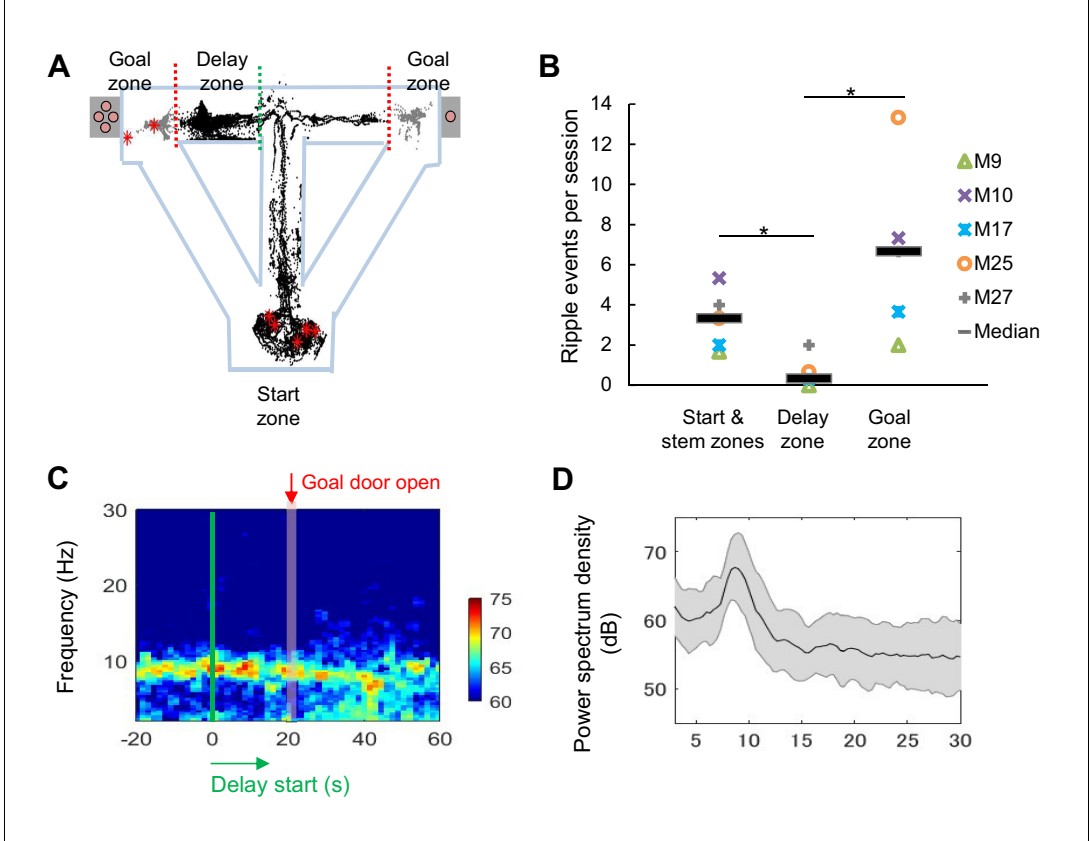

**Figure 2.** LFP signals during the long delay were characterized by strong theta power and lack of SWRs. (**A**) Sharp-wave/ripple events (SWRs) rarely occurred during the delay in the task (data from one session of delay 20 s extension conditions). Red asterisks indicate the locations of SWRs. Black and gray dots indicate the path of animal movements before (black) and after arriving at the goal (gray), respectively. (**B**) SWRs per session in specific experimental zones. The total number of SWRs for each zone was counted and color-coded according to individual animals (the average number of events acquired from 3 sessions of delay 20 s extension conditions). *, p<0.05, Mann-Whitney's *U*-test. (**C**) Spectrogram of the hippocampal CA1 region during the peri-delay period (averaged from three mice, total six sessions of delay 20 s extension conditions) Green line: delay-onset; red line: estimated goal-onset. (**D**) Power spectrum density during 2 s at the beginning of the delay. Shaded area indicates ± SD.

The online version of this article includes the following source data and figure supplement(s) for figure 2:

**Source data 1.** Source Data File for *Figure 2B*.
**Figure supplement 1.** Histological verification of recording sites in the CA1.
**Figure supplement 2.** Cell-type classification by plot in peak-trough and spike width.

large numbers of neurons exhibited significant increases (CA1: 243/639 units: 38.0%) or decreases (CA1: 313/639: 48.9%) in their firing rates during the delay (*Table 1*). We termed these delay-active (delay-act) and delay-suppressed (delay-sup) neurons, respectively (*Figure 4A*). Comparison between the firing rates for short delays (5 s) and those for long delays (20–40 s) revealed that some delay-act and delay-sup cells exhibited significant elevation or reduction of firing rates for specific delay lengths (*Figure 4B*). At the population level, peak firing times of both CA1 delay-act and delay-sup cells were highly distributed across the time spent in the delay zone (*Figure 4C*). To assess

**Table 1.** The number of delay-active and delay-suppressed CA1 excitatory and inhibitory neurons recorded from all sessions.

| Cell type | Delay-active | Delay-suppressed | Other | Total |
|---|---|---|---|---|
| Excitatory neurons | 243 | 313 | 83 | 639 |
| Inhibitory neurons | 43 | 100 | 26 | 169 |

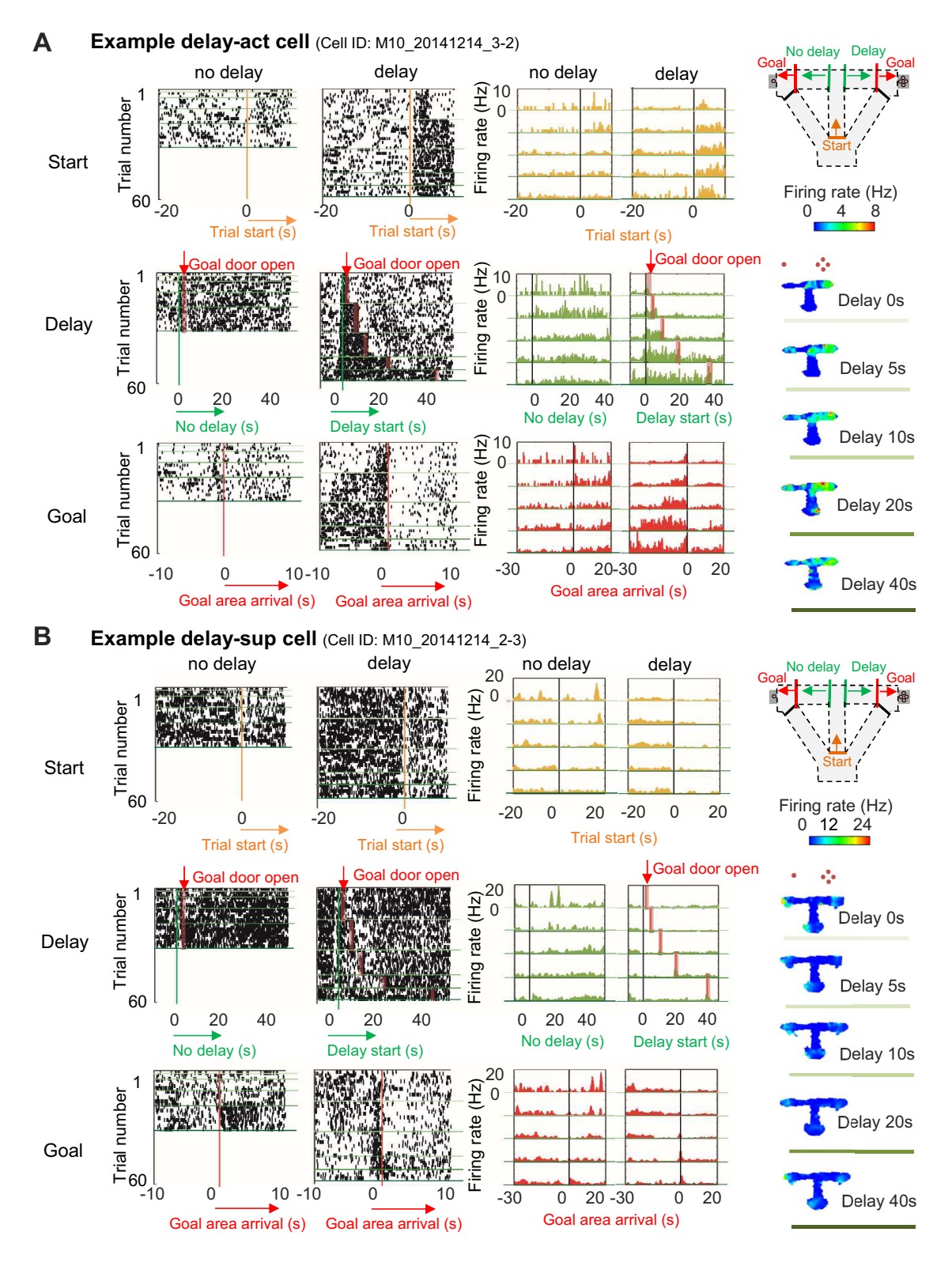

**Figure 3.** Increased or decreased neuronal activity of CA1 cells during delay. (**A**) An example of CA1 delay-active (delay-act) cells, which showed an increment in the firing rate as a function of delay length. Left, raster plots of the firing activity of the cells aligned with start-onset (top), delay-onset (middle) and goal-onset (bottom). Orange lines indicate start-onset. Green lines indicate delay-onset. Pale red lines indicate expected delay-offset. Red lines indicate goal-onset. Center, peristimulus time histograms (PSTHs) of the firing activity of the cells aligned with start-onset (top), delay-onset

*Figure 3 continued on next page*

*Figure 3 continued*

(middle) and goal-onset (bottom). Right, color-coded rate maps. The delayed arm was assigned to the right side with a large reward for this recording session. Red dots indicate the number of sugar pellets. (B) An example of CA1 delay-suppressed cells, which showed a decrement in the firing rate during delay. The delayed arm was assigned to the right side with a large reward for this recording session.

the population activity of CA1 cells during the task, we examined the autocorrelation of the population vectors under long-delay conditions (*Figure 4D*). The population activity of both delay-act and delay-sup cells was clearly segmented into three periods — start, delay and goal — with differential patterns of sustained activity in each. Similar population codes were found in inhibitory CA1 neurons (*Figure 4—figure supplement 1*). We next analyzed the population codes across the individual experimental conditions. We found a uniform-like distribution only under the both-side condition (Kolmogorov–Smirnov test; *Salz et al., 2016*; p=0.46 for the both-side condition; p<0.05 for all other conditions; *Figure 4—figure supplement 2*), while the remaining three protocols found activity biased towards the early part of the delay.

When we examined the activity of neurons sequentially recorded under all possible delays, we found that lengthening the delay dynamically altered activity, with a substantial fraction of units demonstrating a significant correlation between firing rate and delay length (mean firing rate, 20/58 [34.5%], peak firing rate, 46/58 [79.3%] for delay-act; mean firing rate 31/83 [37.3%], peak firing rate, 60/83 [72.3%] for delay-sup; p=0.01, permutation test, for percentages; *Table 2*, *Figure 5*). This indicates that the hippocampus may encode delay length at the level of individual neurons. Peak firing rates may be a better indicator, as mean firing rates at different delay lengths will result in deformative normalization. Further, decoding analysis using support vector machine (SVM) confirmed that the population codes of firing pattern can also predict delay length (classification into five different delay conditions) (*Figure 5—figure supplement 1*; see 'Decoding of delay length from population spike activity' in 'Materials and methods'). Taken together, the hippocampus may encode delay length using dual coding schemes.

Given the learning-dependent development of hippocampal firing during delay period (*Gill et al., 2011*), we also examined the time shift of firing by delay changes. At the beginning of the daily recording session, about 10% (4/33) of delay-act cells initially fired after the animal reached the goal in the 0 s delay condition, then shifted their firing to the delay period once a delay was introduced (*Figure 5—figure supplement 2*). Interestingly, subsequent elimination of the delay did not result in return to goal-related activity (*Figure 5—figure supplement 2A*). When we compared the firing of CA1 delay-act cells under identical short delay trials occurring before and after long-delay trial blocks (*Figure 5—figure supplement 2B*), we found about 10% of neurons shifted positively or negatively (*Figure 5—figure supplement 2C*), indicating that the onset of firing in the CA1 neurons was influenced by the experience of waiting and/or learning of the delay.

## Place-specific delay information is encoded in the majority of CA1 neurons

Given the robust place code present in the hippocampus, we next asked whether CA1 delay-act neurons were spatially selective. To this end, we switched the location of the delay and no-delay arms (switched conditions) or replicated the delay on the other side (both-side conditions) (*Figure 6A*, *Figure 6—figure supplement 1*), with corresponding changes in reward size. Under both conditions, the mice changed their behavior within several trials, with the preference for the large reward arm reaching about 70%. We then evaluated side-selectivity of the delay activity, adding location as a variable under three-way ANOVA (side, delay-length, and timing; see 'Materials and methods') during switched and both-side trials (*Figure 6A*). Representative side-selective and -unselective excitatory neurons in the CA1 are shown in *Figure 6B*. The percentage of side-selective neurons was high in both CA1 delay-act (114/155: 73.5%) and delay-sup(124/191: 64.9%) neurons (*Figure 6C* and *Tables 2* and *3*), however more than a quarter of the neurons of both groups encoded delay independent of location.

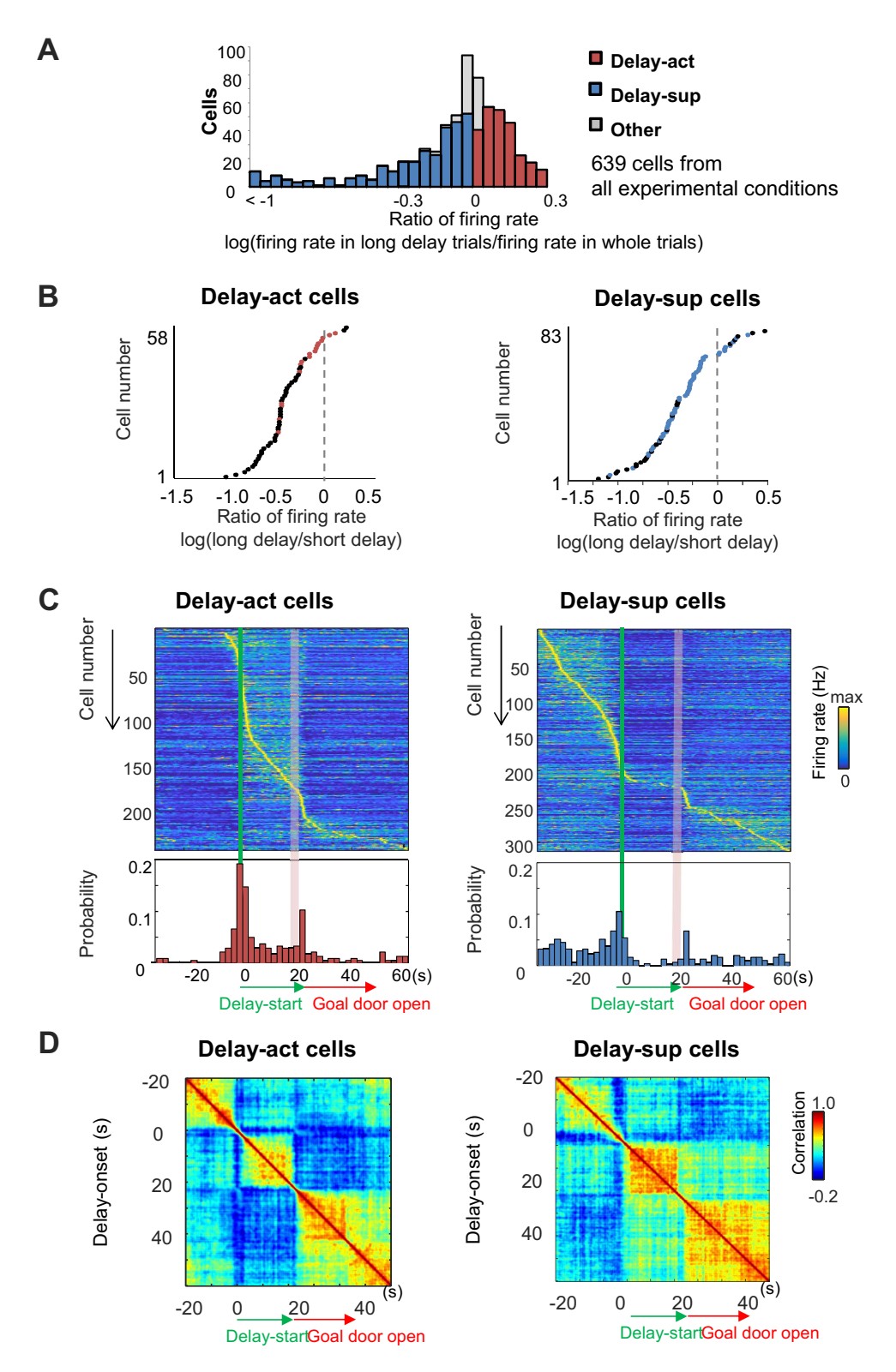

**Figure 4.** Delay-dependent firing patterns of CA1 delay-active and delay-suppressed cells. (**A**) The distribution of delay-active and delay-suppressed cells aligned with the ratio of firing rate in the long delay period and in whole trials. (**B**) The ratio of mean firing rate during long delays (20 or 40 s) to that during short delay (5 s) for all neurons (base-10 log-transformed). Each dot indicates an individual neuron. Black dots indicate neurons that had a statistically significant difference in firing rate between short and long delay conditions (p<0.001). (**C**) Temporal patterns of firing rates in CA1 delay-

*Figure 4 continued on next page*

*Figure 4 continued*

active and delay-suppressed cells during delay. Top, color-coded temporal firing patterns. Neurons were ordered by the time of their peak firing rates. Bottom, temporal distribution of the peak firing rates of the neurons. Green lines indicate delay-onset. Pale red lines indicate expected delay-offset. (D) Correlation matrix of population vectors as a function of time for CA1 delay-active and delay-suppressed cells.

The online version of this article includes the following figure supplement(s) for figure 4:

**Figure supplement 1.** Population coding of long delay in the inhibitory CA1 cells.
**Figure supplement 2.** Population coding of long delay in CA1 delay-active cells in different experimental conditions.

## Value-coding in CA1 neurons

We next asked how subjective value influenced the activity of the delay-act and delay-sup neurons in CA1. As mentioned above, delay and reward are common factors that modulate subjective value. To examine whether the changes of delay period firing patterns in delay increment were correlated with changes in reward size, we first lengthened the delay length and subsequently decreased the reward for the delayed option (reward loss conditions) and followed this by restoration of reward. In addition, we manipulated the reward size in the opposite direction to avoid order-dependent confounds arising from decreased hunger or motivation of the animals in later trials (reward gain conditions; *Figure 7A*, *Figure 7—figure supplement 1*). The majority of delay-act cells decreased their firing rate in response to reward loss, whereas delay-sup neurons had the opposite response, increasing their activity (*Figure 7B and C*). As a result, the log ratio of the firing rates (large reward/small reward) under both reward loss and gain conditions was significantly different between delay-act and delay-sup cells ($Z = -2.6$, p=0.007 for reward loss; $Z = 2.1$, p=0.03 for reward gain, Mann-Whitney's *U*-test). In total, the ratio was negatively skewed in the delay-act cells ($T = -2.5$, p=0.01, one-sample *t*-test) but positively skewed in delay-sup cells ($T = 2.7$, p=0.01, one-sample *t*-test, *Figure 7D*). These results suggest that firing during the delay independently reflected positive and negative outcomes in these different subpopulations of CA1 neurons. Finally, we examined the relation between firing rate changes 'by delay extension' and 'by reward manipulations' to explore whether value, a more general concept of information, may be neutrally encoded. In both delay-act and delay-sup cells, there was no global trend, but a subset of neurons, plotted around the line of 'delay effect = reward effect' (*Figure 7E*), can be interpreted as value-coding neurons.

We next focused on the relationship between the behavioral shift during reward loss sessions and the firing patterns of delay-act cells. If CA1 activity is dependent on the animals' choice preference, the activity should be dynamically changed after the elimination of preference. However, across the session, animals avoided the delayed option, making it difficult to observe CA1 activity under this condition. To eliminate the preference for the delayed options, we designed an 'unequal conditions' (long delay + no pellet vs long delay + four pellets, with the latter being the better option). Animals then quickly reduced their preference to the delayed option with no reward. To record the activity for the less-preferred or adverse choice, we forced mice to choose the less-preferred option with an obstacle set at the entrance of the opposite arm. When faced with an unrewarded delayed option, CA1 neurons indicating choice preference were silent (*Figure 7—figure supplement 2*). These

**Table 2.** Full distribution of CA1 excitatory neurons for all of the tested conditions.

| Test conditions | Delay responsiveness | Neuron number |
|---|---|---|
| Extension | Delay-active | 58 |
| | Delay-suppressed | 83 |
| | Other | 36 |
| Switched or both-side | Delay-active | 155 |
| | Delay-suppressed | 191 |
| | Other | 34 |
| Reward loss or gain | Delay-active | 30 |
| | Delay-suppressed | 39 |
| | Other | 13 |

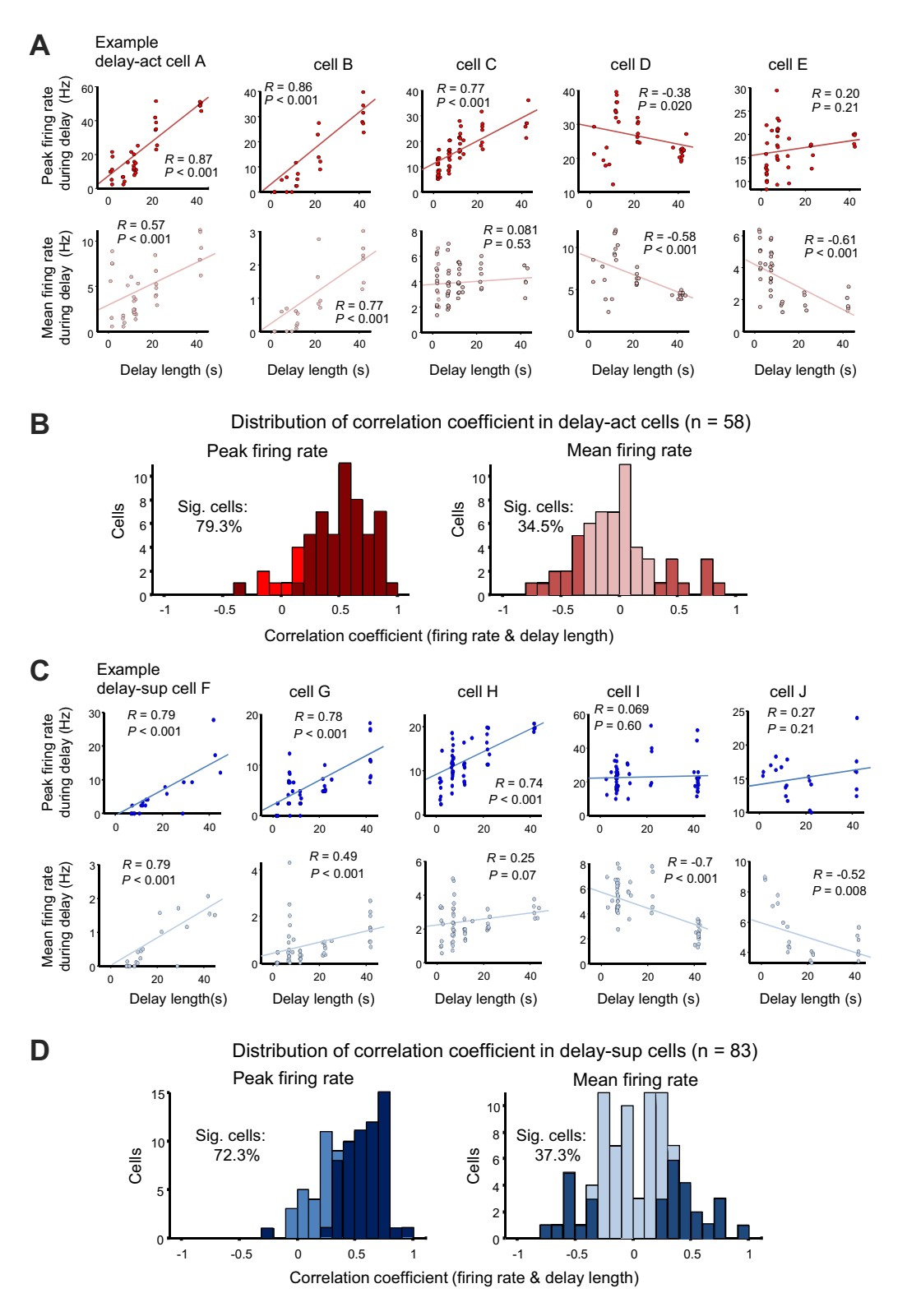

**Figure 5.** Delay-dependent firing patterns of CA1 delay-active and delay-suppressed cells. (**A**) Scatter plots show correlations between firing rate during delay (upper, peak firing rate; lower, mean firing rate) and the delay length of five representative delay-act cells. Cells A and B show positive correlations in both peak and mean firing rate; on the other hand, cell C shows either. Cells D and E show negative correlations. (**B**) Distribution of correlation coefficients between firing rate (left, peak firing rate; right, mean firing rate) and delay length in delay-act cells. Dark color bars indicate

*Figure 5 continued on next page*

*Figure 5 continued*

statistically significant neurons (p<0.05), whereas bright color bars indicate neurons do not reach statistical significance. (**C**) Scatter plots show correlations between firing rate during delay (upper, peak firing rate; lower, mean firing rate) and delay length for five representative delay-suppressed cells. (**D**) Distribution of correlation coefficients between firing rate (left, peak firing rate; right, mean firing rate) and delay length in delay-suppressed cells. Dark color bars indicate statistically significant neurons (p<0.05), whereas bright color bars indicate that neurons do not reach statistical significance.

The online version of this article includes the following figure supplement(s) for figure 5:

**Figure supplement 1.** Decoding of delay length from population spike activity.
**Figure supplement 2.** Temporal patterns of the CA1 neuron shift as sessions progressed.
**Figure supplement 3.** Population coding of long delay in the delay-length-correlated and non-correlated CA1 cells.

results suggest that the firing of delay-act neurons in the CA1 region represents the animal's subjective value of the chosen options.

## NMDAR deficiency in hippocampus disrupted delay-discounting and populational delay coding in CA1

Finally, we took advantage of a mutant mouse, the *CaMK2-Cre; NR1-flox/flox* mouse, which lacks CA1 pyramidal cell N-methyl-D-aspartate (NMDA) receptors (NMDARs) (CA1-NR1cKO mouse; *McHugh et al., 1996*; *Tsien et al., 1996a*), RRID:MGI:3581524), to assess the role of synaptic plasticity in task performance. Consistent with previous reports of hippocampus-dependent learning deficits in these mice (*Bannerman et al., 2012*; *Rondi-Reig et al., 2001*; *Tsien et al., 1996b*), they exhibited impaired delay discounting (*Figure 8A*), demonstrating a significant bias for the larger reward even when the delay was extended ($F_1$ = 14.4, p<0.001, genotype (CA1-NR1cKO vs NR1 f/f); $F_4$ = 23.0, p<0.001, interaction between delay length x genotype, $F_{1,4}$ = 1.07, p=0.37, two-way ANOVA, p=0.61 on delay 0 s, p=0.04 on delay 5 s, p=0.04 on delay 10 s, p=0.002 on delay 20 s, p=0.005 on delay 40 s, multiple comparisons on each delay length).

We next recorded CA1 neuronal activity in cKO (n = 3, 123 units) and control mice (n = 4, 69 units, *Table 4* and *Table 5*) to look for physiological correlates of the behavioral change. Delay-act cells in the cKO mice showed non-specific activation during the delay period (*Figure 8B* and *Figure 8—figure supplement 1A*). Hence, there was a lower and higher proportion of delay-act and delay-sup neurons, respectively, in the cKO and the ratio of delay-act/delay-sup was significantly lower in the cKO than in the control mice (*Figure 8C*, p=0.02, Fisher's Exact Test, *Figure 8—figure supplement 1B*). Further, in contrast to the controls, the temporal distribution of all delay-act cells in the cKO was sparse and not specific to delay-onset. As a result, population vector analysis revealed that the activity was not segmented into three periods in the cKO mice (*Figure 8D*). In addition, the ratio of CA1 firing of cKO was significantly different than that observed in control mice and lacked the expected negatively skewed distribution (Z = 2.0, p=0.04, Mann-Whitney's *U*-Test, *Figure 8E*). We could not detect significant difference among the genotypes in basic firing property during the task (mean firing rate, cKO — 3.07 Hz, control — 3.39 Hz, Z = −0.76, p=0.44, Mann-Whitney's U-test). Subpopulation firing rates were also not significantly different (delay-act, cKO — 2.66 Hz, control — 3.13 Hz, Z = −0.91, p=0.35; delay-sup, cKO — 3.47 Hz, control — 3.47 Hz, Z = −0.68, p=0.14, Mann-Whitney's *U*-test). These findings suggest that delay discount behavior and the underlying delay-related activity in CA1 pyramidal cells requires NMDAR-dependent mechanisms in the hippocampus.

## Discussion

We recorded CA1 neuronal activity in mice during delay-based decision making in an automated T-maze task while independently manipulating delay length and reward size across sessions. We observed distinct populations of neurons that increased or decreased their firing during the delay. Moreover, the firing rates of a subset of the delay-activated CA1 neurons decreased with both delay length increments and reward size declines. Notably, the activated and suppressed neurons showed distinct activity changes following reward size manipulations. These results suggest that dissociable subpopulations of hippocampal neurons represent delay and reward information in opposing ways. These discoveries should help shape models of how the hippocampus supports decision making.

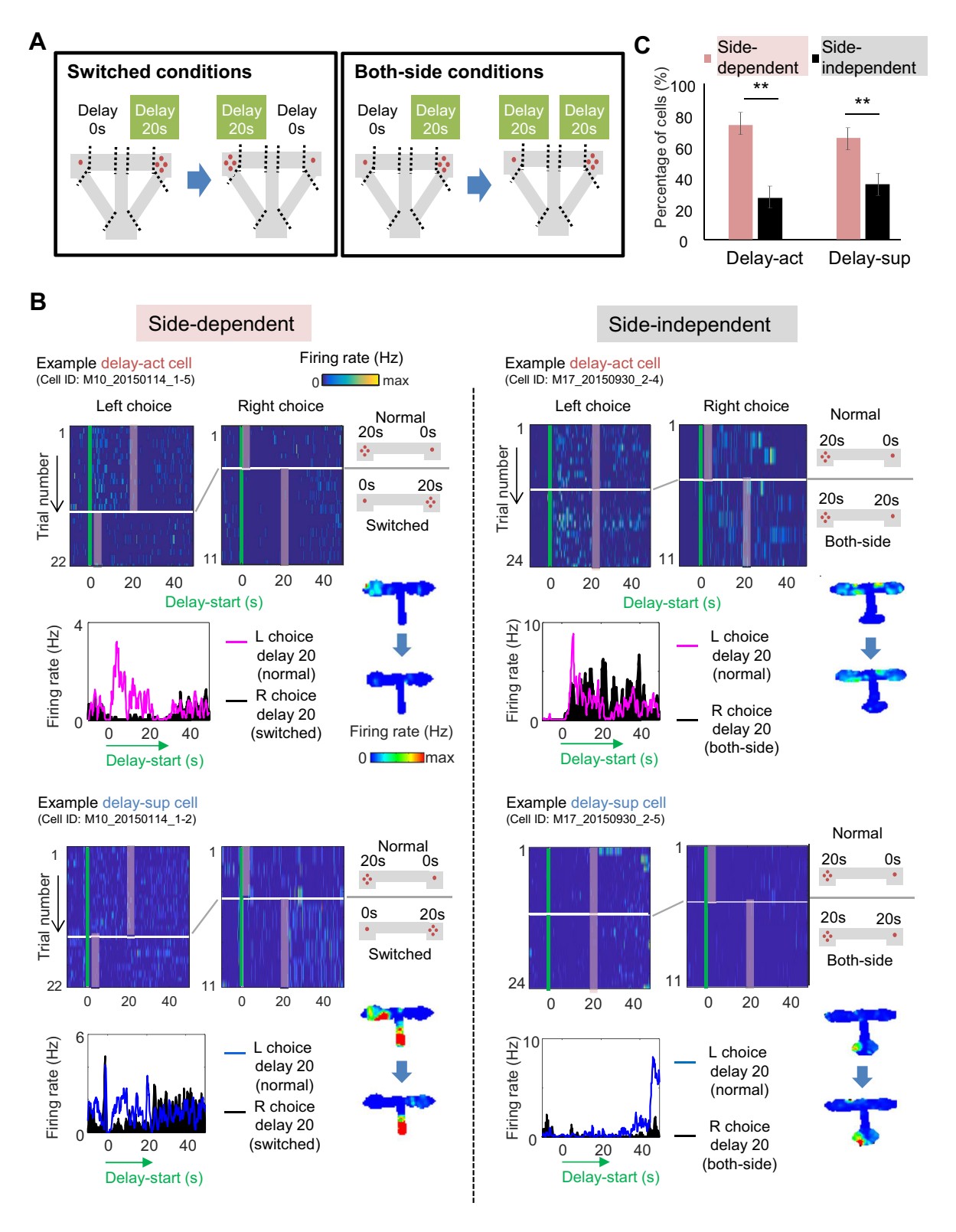

**Figure 6.** Spatial-selective delay coding in CA neurons. (**A**) Experimental conditions to investigate the location selectivity in delay-active neurons. The location of the delay zone was switched to the other side (switched conditions) or doubled to both sides (both-side conditions). (**B**) Example CA1 delay-active and delay-suppressed cells. Side-dependent and side-independent neurons are shown as left and right rows, respectively. Top left, colored raster plots expressing relative firing rates. Green lines indicate delay-onset. Pale red lines indicate expected delay-offset. Top right, information of

*Figure 6 continued on next page*

*Figure 6 continued*

conditions corresponded to the raster plots on the left. Red dots indicate the number of sugar pellets. Bottom left, Peri-event time histograms showing the averaged firing rates. Magenta lines indicate the firing rate of the left choice with a 20 s delay. Black-filled histograms indicate the firing rate of the right choice with a 20 s delay. Bottom right, color-coded rate maps for the two conditions (normal delay and switch or both-side conditions). (C) Percentage of place-dependent and -independent CA1 delay-active and delay-suppressed neurons. Error bars indicate 95% Clopper-Pearson's confidence intervals. **: p<0.01, Mann-Whitney's *U*-test.

The online version of this article includes the following figure supplement(s) for figure 6:

**Figure supplement 1.** Daily timeline of the switch and both-side conditions.

Although the delay-modulated activity was diverse across CA1 neurons, their responsiveness to delay was precisely controlled. A significant fraction of CA1 neurons reflected delay length in their firing rate, suggesting the encoding of delay length in the hippocampus on a single-cell level. Related to this, positive and negative correlations with delay length were observed in both delay-act and delay-sup cells. Currently, it is not clear what roles delay-act or delay-sup cells or those neurons with positive or negative correlation play in the animal's decision. In a delay-discounting task, delay may be encoded in two different ways: by a discounting factor and by a factor predicting a larger reward. Future work should investigate whether the two directions of correlation are related to the discounting or prediction.

At the population level, the peak firing rates of delay-act and delay-sup cells were distributed largely around the delay-onset. As a result, population vector analysis demonstrated segmented and sustained network activity during the delay in the CA1 region, suggesting a role in prospective coding of specific periodic events centered on the delay. The decoding analysis demonstrated that particularly during short delay blocks, delay length could be decoded with population activity even prior to the delay initiation. This may reflect the animal's experience with the task and expectation of an impending reward. In addition, in the specific circumstance where a fixed delay was constantly presented, the population coding of delay may be more precise.

A significant fraction of both the delay-act and the delay-sup neurons that we recorded also carried spatially tuned delayed information. Thus, the activity of most delay-act and delay-sup cells in dorsal CA1 does not appear to represent solely delay information, but rather, may represent integrated information of the chosen option, reflecting both location and delay. This result is consistent with the idea that hippocampal cells are coding not only within the space and time dimensions individually, but rather across them jointly (*Eichenbaum, 2014*; *Howard and Eichenbaum, 2015*; *MacDonald et al., 2011*).

Changing reward size modulated the firing rates of both the delay-act and the delay-sup cells in CA1. It is widely known that the activity of CA1 neurons can depend on reward (*Ambrose et al., 2016*; *Hölscher et al., 2003*; *Singer and Frank, 2009*). Studies focusing on goal-directed behavior have demonstrated that some CA1 neurons fire when animals approach, wait for, or acquire rewards, but not when animals visit the same location in the absence of the reward (*Eichenbaum et al., 1987*; *Fyhn et al., 2002*; *Hok et al., 2007*; *Kobayashi et al., 2003*; *Rolls and*

**Table 3.** Distribution of side-dependent and side-independent, delay-active and delay-suppressed, CA1 excitatory and inhibitory neurons.

| Delay responsibility | Side-dependency | Cell types | N | % |
|---|---|---|---|---|
| Delay-active | Side-dependent | Excitatory neuron | 114 | 73.5 |
| | | Inhibitory neuron | 12 | 60.0 |
| | Side-independent | Excitatory neuron | 41 | 26.5 |
| | | Inhibitory neuron | 8 | 40.0 |
| Delay-suppressed | Side-dependent | Excitatory neuron | 124 | 64.9 |
| | | Inhibitory neuron | 45 | 71.4 |
| | Side-independent | Excitatory neuron | 67 | 35.1 |
| | | Inhibitory neuron | 18 | 28.6 |

*Xiang, 2005*), indicating that a certain subset of CA1 neurons are highly sensitive to reward expectation or motivation. However, in monkeys, omission of a reward activated some CA1 neurons (*Watanabe and Niki, 1985*). This is consistent with our results demonstrating that during the delay, dissociable subsets of CA1 neurons were positively or negatively correlated with reward size. The scatter plot of the firing rate ratio of small/large reward conditions and long/short delay conditions (*Figure 7E*) shows that there are no global trends, suggesting that the CA1 neurons exhibit independent relationship between delay and reward manipulation responses. We found, however, that a fraction of neurons reacted in the same way to delay and reward manipulation, suggesting that there may be value-coding neurons in the CA1. Further study will be required to isolate specific neurons encoding subjective value, focusing on specific pathways or cell types. Accordingly, a distinct subpopulation of CA1 neurons may encode the delay-reward integration and may support the valuation process in delay-based decision making.

The phenotype of 'lowered delay discounting' caused by a loss of the NMDAR may also be interpreted as an abnormal repetition of an unpleasant choice, referred to as 'compulsive behavior'. Systemic injection of the partial NMDAR agonist D-cycloserine reduces compulsive lever-pressing in a model of obsessive-compulsive disorder (OCD) in rats (*Albelda et al., 2010*). In addition, polymorphisms in a subunit of NMDAR have been considered as a risk factor in OCD (*Arnold et al., 2004*). The present study suggests that the hippocampal NMDARs are required for delay discounting and provides additional evidence that hippocampal NMDARs may be associated with compulsive disorders. It is widely believed that synaptic plasticity via NMDAR-dependent machinery contributes to association learning and that, in the hippocampus, this contributes to the formation of long-term, spatial memories (*Martin et al., 2000*). Studies using several lines of conditional knockout mice have pointed out that NMDAR in the hippocampus is involved in spatial learning (*Tsien et al., 1996a*), nonspatial learning (*Huerta et al., 2000*; *Rondi-Reig et al., 2001*), anxiety (*Bannerman et al., 2004*; *Kjelstrup et al., 2002*; *McHugh et al., 2004*; *Richmond et al., 1999*), time perception (*Huerta et al., 2000*), and decision making (*Bannerman et al., 2012*). In addition, physiological studies have demonstrated that a hippocampus that lacks NMDAR exhibits less specific spatial representation in place cells (*McHugh et al., 1996*). We found that NMDAR deficiency disrupted the proportion of delay-act and delay-sup cells, and population coding for the delay. These findings suggest that the NMDAR in the hippocampus may be required to maintain or develop time-coding. It should be noted that the NR1 knockout may be extended to other telencephalic regions (CA3, dentate gyrus, deep cortical layers) in the cKO animals in the present study. Further research is required in order to identify more specific mechanistic roles of the hippocampal NMDAR in delay-based decision making. In addition, in contrast to previous studies showing that rats with hippocampal lesions exhibit higher discount rates (*Cheung and Cardinal, 2005*; *Mariano et al., 2009*; *McHugh et al., 2008*), the NMDA KO mice demonstrated the opposite phenotype. Thus, there may be considerable differences between the effect of the lesions and that of NMDAR knockout in the hippocampus on the full network engaged during delay-based decision making.

In conclusion, our results show that CA1 neuronal activity during delay is segregated into two populations, delay active and delay suppressed neurons. Further, these groups demonstrate opposing responses to changes in motivational background. In addition, NMDAR-dependent plasticity mechanisms appear to be required for the formation of the firing patterns during delay and for the delay-discounting. These findings further clarify the role of the hippocampus in decision making, as well as in the control of impulsive or compulsive behaviors.

## Materials and methods

### Key resources table

| Reagent type (species) or resource | Designation | Source or reference | Identifiers | Additional information |
|---|---|---|---|---|
| Strain, strain background (*Mus musculus*) | C57BL/6J | RIKEN Bio Resource Center | RRID: IMSR_JAX:000664 | Wild-type mouse |

*Continued on next page*

*Continued*

| Reagent type (species) or resource | Designation | Source or reference | Identifiers | Additional information |
|---|---|---|---|---|
| Strain, strain background (*Mus musculus*) | NR1$^{flox}$ | PMID: 8980237 | 005246 (Jackson Laboratory) | Targeted mutation line |
| Strain, strain background (*Mus musculus*) | Tg(Camk2a-cre) T29-1Stl/J | PMID: 8980237 | 005359 (Jackson Laboratory) | Cre transgenic line |
| Strain, strain background (*Mus musculus*) | Tg(Camk2a-cre) T29-1Stl/J, NR1$^{flox/flox}$ | PMID: 8980237 | RRID: MGI:3581524 | Conditional knockout line |
| Commercial assay or kit | T-maze | O'hara and Co., Ltd. | RRID: SCR_018016 | Automatic operant test |
| Other | Neural probes | NeuroNexus | A4 × 2-tet-5mm -150-200-312 | 32-ch electrode |
| Other | nDrive | NeuroNexus | RRID: SCR_018019 | Micro driver to control movement of electrode |
| Other | Amplipex: KJE-1001 | Amplipex | RRID: SCR_018017 | Recording system for neural signals |
| Software, algorithm | MATLAB_R2018a | Mathworks | RRID: SCR_001622 | |
| Software, algorithm | Klusters | PMID: 16580733 | RRID:SCR_015533 | |
| Software, algorithm | NDmanager | PMID: 16580733 | RRID:SCR_015533 | |
| Software, algorithm | Neuroscope | PMID: 16580733 | RRID:SCR_015533 | |
| Software, algorithm | KlustaKwik2 | PMID: 25149694 | RRID:SCR_014480 | |
| Sequence-based reagent | Cre_F | PMID: 28244984 | PCR primers | ACC TGA TGG ACA TGT TCA GGG ATC G |
| Sequenced-based reagent | Cre_R | PMID: 28244984 | PCR primers | TCC GGT TAT TCA ACT TGC ACC ATG C |
| Sequenced-based reagent | NR1$^{flox}$-F | PMID: 28244984 | PCR primers | TGT GCT GGG TGT GAG GGT TG |
| Sequenced-based reagent | NR1$^{flox}$-R | PMID: 28244984 | PCR primers | GTG AGC TGC ACT TCC AGA AG |
| Other | DAPI stain | ThermoFisher | Thermo Fisher Scientific Cat# D1306 | (1 µg/mL) |
| Other | DiI stain | ThermoFisher | Thermo Fisher Scientific Cat# D3911 | (200 µg/mL) |

## Animals

All procedures were approved by the RIKEN Animal Care and Use Committee. A total of 29 male C57B6/J mice were used for this study (wildtype, n = 5; cKO, n = 11 (8 for behavioral study); control: n = 13 (9 for behavioral study]). Mice lacking NMDAR in the hippocampus (RRID:MGI:3581524) were generated by crossing the line gene-targeted for loxP-tagged *Nr1* (*Grin1*) alleles (*Nr1*$^{flox}$; *Tsien et al., 1996a*) and a transgenic line carrying *Camk2a* promoter-driven Cre recombinase (*Camk2a-Cre*, T29-1Stl; *Tsien et al., 1996a*). In this mutant, deletion of NR1 is delayed until about 4 weeks after birth and is restricted to the CA1 pyramidal cells until about 2 months of age (*Fukaya et al., 2003*). Most of the behavioral analysis using the mutant was done until this age. Hence, it is unlikely that the behavioral impairment observed was the result of undetected developmental abnormalities. Physiological characterization, however, may have harbored a more

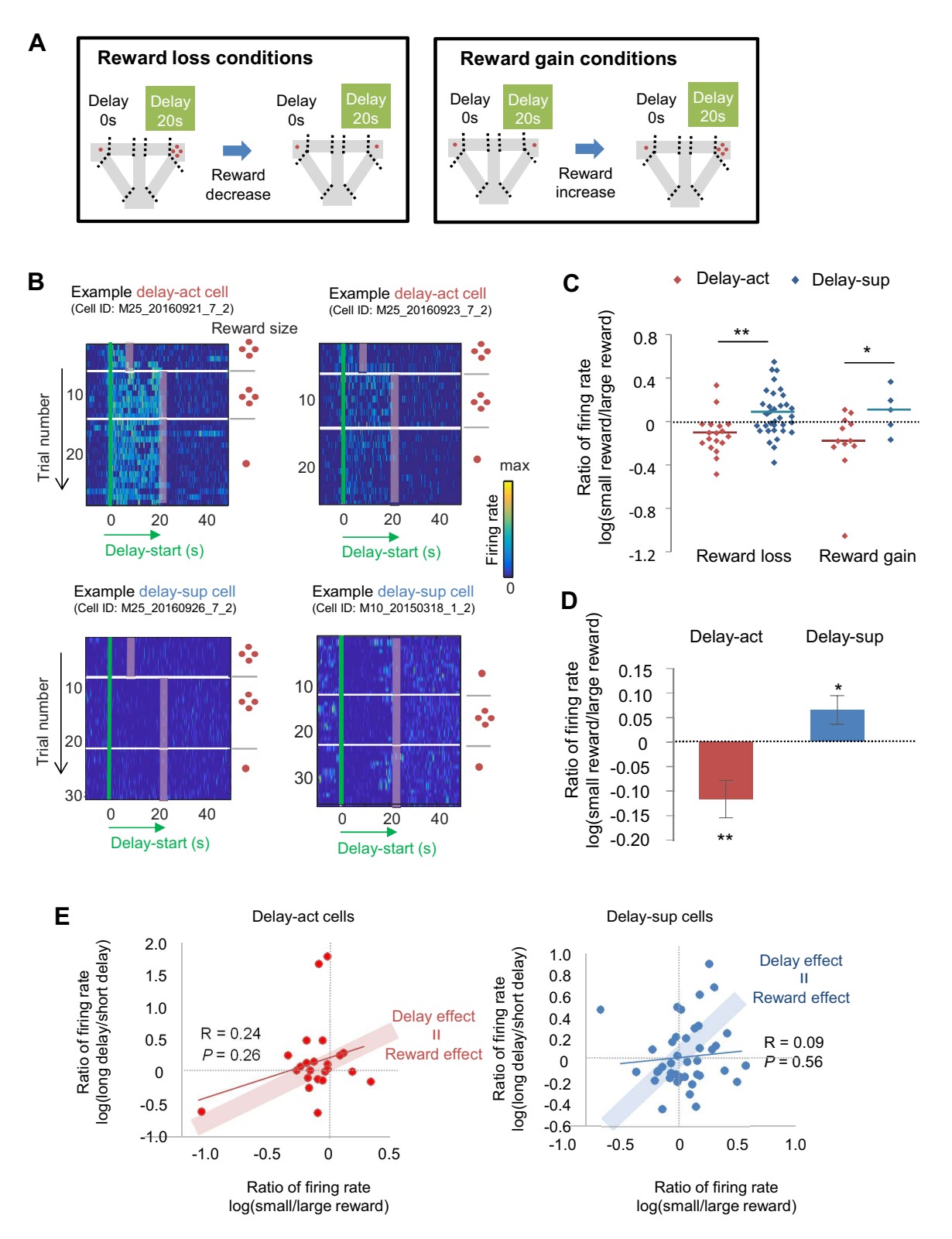

**Figure 7.** The firing of CA1 delay-active and delay-suppressed cells is distinctly changed by reward size manipulations. (**A**) Left, experimental reward loss conditions: the reward size was changed from 4 to 1 (or 0) pellets. Right, experimental reward gain conditions: the reward size was changed from 1 (or 0) to 4 pellets. (**B**) Example CA1 delay-active (top) and delay-suppressed cells (bottom) fired during delay in reward loss conditions. Green lines indicate delay-onset. Red lines indicate expected delay-offset. Red dots indicate the number of sugar pellets. (**C**) Ratio of firing rates of delay-active

*Figure 7 continued on next page*

*Figure 7 continued*

and -suppressed cells in reward loss and gain conditions. Dots indicate individual data for delay-active cells (red) and delay-suppressed cells (blue). Central bars indicate the medians. *, p<0.05; **, p<0.01, Mann-Whitney's *U*-test. (D) Ratio of firing rates of delay-active and -suppressed cells in mixed population. Error bars indicate SEM. *, p<0.05; **, p<0.01, One-sample *t*-test. (E) Scatter plots of firing rate ratios between small/large reward conditions and between long delay/short delay conditions. The computed correlation coefficient R and p value are indicated.

The online version of this article includes the following source data and figure supplement(s) for figure 7:

**Source data 1.** Source Data File for *Figure 7C and D*.
**Figure supplement 1.** Daily timeline of reward loss and gain conditions.
**Figure supplement 2.** Firing in CA1 delay-active cells depended on animal preference.

widespread deletion of the NR1 gene as the ages of cKO animals in the recording session were slightly more than 2 months old (*Table 5*).

## Delay-based decision-making task

Adult mice were trained in a delay-based decision-making task under an automated T-maze (O'HARA and Co., Tokyo, Japan, RRID:SCR_018016) before electrophysiological recording. The maze was partitioned off into six areas (Start, Junction, Right-Goal, Right-Back, Left-Goal, and Left-Back) by seven sliding doors (S-J, J-R, R-RG, RG-S, J-L, L-LG, and LG-S). The detailed protocol has been described previously (*Kobayashi et al., 2013*; *Zhang et al., 2018*). In short, the mice had food restriction to approximately 80% of free-feeding weight, were habituated to the maze, and baited with scattered pellets (30 min/day) for 2 days. The large reward arm and the small reward arm were allocated to the right or left side arm randomly for each mouse. Four pellets were available in the large reward arm, whereas only one pellet was available in the small reward arm. Mice were allowed to roam freely and without delay to select either arm for 5–10 days for the initial training period until they preferred the large arm (>80%). Then, all animals were trained in the extension delay conditions for at least 5 days. For the first block of trials for each day, the large reward arm was associated without delay (0 s), and then, during the later blocks, it was associated with a 5 s, 10 s, 20 s, or 40 s delay. In the meantime, the small reward arm was always associated with no delay. Each block consisted of 10 trials or more (15 or 30 min). If the trial number was lower than 10, additional blocks were employed. Next, the mice, except cKO and control, were trained in the switched and both-side conditions. In the switched condition, the side of the delayed-large arm was switched to the other side. In the both-side condition, both sides were set as delayed-small and delayed–large arms. The switched conditions were performed initially and then under both-side conditions. In changing the conditions, 10 or more trials were continuously performed to develop a sustained reaction from the animals. Finally, the mice were trained in the Reward loss and gain conditions. We decreased the reward size to investigate whether the firing rate reflected a positive or negative aspect in the delayed option. Initially, we set a delay for a short time with the normal large reward, and then we changed the delay to be long without any change in the large reward, similar to other conditions. After these two continuous sessions, we changed the reward size from four to one pellet. As for other control conditions, we also performed the opposite flow (long delay with one pellet first, long delay with four pellets next). For all experiments, during the time between blocks, mice were allowed to drink water. Four to six consecutive daily sessions were performed per week.

## Histological identification of the localization of the recorded sites

Owing to the small thickness of the silicon probe shanks, the tracks of shanks were hard to detect. Painting at the back of the shanks with DiI (Thermo Fisher Scientific Cat# D3911) and/or the creation of an electrical lesion by a small current (5 mA for 5 s) was used to facilitate track identifications under DAPI staining (Thermo Fisher Scientific Cat# D1306) (*Figure 2—figure supplement 1*).

## Recording and spike sorting

Mice were anesthetized with isoflurane during surgery. Silicon probes or wire tetrodes were implanted in the hippocampal CA1 region (AP = −2.0 to −2.8 mm, ML = 1.2 to 2.0 mm, DV = 1.2 to

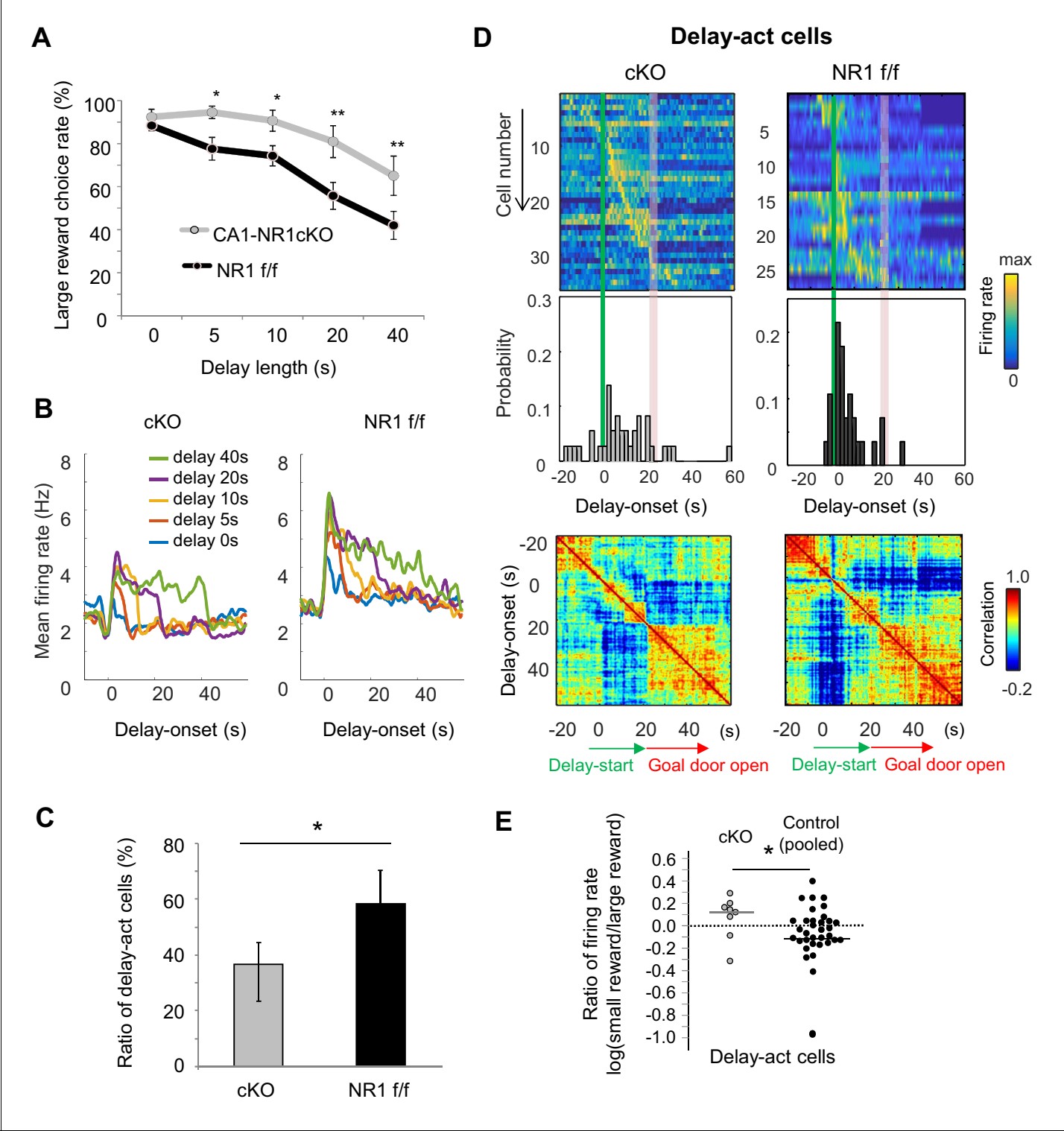

**Figure 8.** NMDAR-dependent mechanism for delay-discounting. (**A**) Impaired delay-discounting in CA1-NR1 cKO mice. *, p<0.05; **, p<0.01; post-hoc Scheffe's test. Error bars indicate SEM. (**B**) NMDAR deficiency disrupted the delay tuning in the CA1 activity. Average firing patterns of the CA1 delay-active cells from cKO and control mice for different delay lengths (0, 5, 10, 20, and 40 s). (**C**) Abnormal delay-active and –suppressed cell proportion in cKO mice. Ratio of delay-act cells to delay-sup cells for cKO and control mice. Error bars indicate 95% Clopper-Pearson's confidence intervals. *, p<0.05; Mann–Whitney's *U* test. (**D**) NMDAR deficiency disrupted the populational activity in CA1. Top, color-coded temporal firing patterns of the CA1 delay-active cells in cKO and control mice. Neurons were ordered by the time of their peak firing rates. Middle, temporal distribution of neurons. Green lines indicate delay-onset. Red lines indicate expected delay-offset. Bottom, correlation matrix of population vectors as a function of time for CA1

*Figure 8 continued on next page*

*Figure 8 continued*

delay-act cells in cKO and control mice. (E) NMDAR deficiency disrupted the negative skew in the firing rate ratio of delay-active cells. Ratio of firing rates of delay-active cells in CA1 of cKO and WT mice. Dots indicate individual data for cKO (gray) and control (black) mice. The central bar indicates the median. *, p<0.05; Mann–Whitney' *U* test.

The online version of this article includes the following source data and figure supplement(s) for figure 8:

**Source data 1.** Source Data File for *Figure 8A and E*.

**Figure supplement 1.** Mutant mice (CA1-NR1cKO) exhibiting impaired delay discounting showed less specific spatial representation in place cell activities and less in the number of delay-active cells.

---

1.5 mm). In all experiments, ground and reference screws were fixed in the skull atop the cerebellum. The silicon probes attached to micromanipulators (nDrive, NeuroNexus, Michigan, USA), or to nichrome wire tetrodes combined with a micro-drive (*Middleton and McHugh, 2016*), which enabled us to move their positions to the desired depth, were implanted into the mice. Electrophysiological signals were acquired continuously at 20 kHz on a multi-channel recording system (KJE-1001, Ampliplex Ltd, Szeged, Hungary, RRID:SCR_018017). The wide-band signal was down-sampled to 1.25 kHz and used as the LFP signal. We detected SWRs (their timing, power, and durations) from filtered signal (120–230 Hz), which corresponded to more than three SD of log-power in the same frequency band. To trace the temporal positions of the animals, two color LEDs were set on the headstage and were recorded using a digital video camera at 30 frames/s. Spikes were extracted from the high-pass filtered signals (median filter, cut-off frequency: 800 Hz). Spike sorting was performed semi-automatically, using KlustaKwik2 (RRID:SCR_014480, https://github.com/kwikteam/klustakwik2/; *Kadir et al., 2014*). The cell types of the units were classified by peak-trough latency and width. In total, we analyzed 831 putative excitatory neurons (n = 639 for wildtype; n = 123 for cKO; n = 69 for NR1f/f mice; *Table 1* and S4) and 250 inhibitory neurons (n = 169 for wildtype; n = 53 for cKO; n = 28 for NR1f/f mice). The positions of the animals were determined by the position of the LEDs mounted on the headstage. The rate maps of the spike number and occupancy probability were generated from 4 cm binned segments from the position and spiking data. The normalized PSTH for individual neurons in delay-act and delay–sup cells in the CA1 was computed under delay 20 s conditions. The autocorrelation of the population vector was then computed.

## Determination of delay-active and delay-suppressed neurons

To examine the effect of delay on neuronal activities, we quantified changes in the firing rate of each neuron during the long delay period. First, we calculated the firing rate in the delay zone ($R_{delay}$ = spike number in delay zone $n_{delay}$/time spent in delay zone $t_{delay}$; see *Figure 1A*), and that in all zones ($R_{total}$ = spike number in all zones $n_{total}$/time spent in all zones $t_{total}$) in long-delay trials, and then computed the ratio of them ($R_{delay}/R_{total}$). Second, we performed a permutation test in order to determine whether the ratio of the firing rates $R_{delay}/R_{total}$ shows significant change or not. To make surrogate data, we resampled the spike trains by permuting the inter-spike-intervals and by realigning with them. We repeated this process 1000 times to obtain 1000 resampled datasets. The rank of the original firing rate ratio $R_{delay}/R_{total}$ in the resampled 1000 firing rate ratios was used to

---

**Table 4.** Full distribution of CA1 excitatory neurons for the NMDAR mutant study.
The numbers in parentheses are cells from the wildtype.

| Test conditions | Delay responsiveness | Neurons | |
|---|---|---|---|
| | | cKO | Control |
| Extension | Delay-active | 28 | 25 |
| | Delay-suppressed | 56 | 20 |
| | Other | 22 | 19 |
| Reward loss and gain | Delay-active | 8 | 33 (30) |
| | Delay-suppressed | 6 | 0 |
| | Other | 3 | 2 |

**Table 5.** Ages of CA1-NMDAR cKO mutant and control mice used for the electrophysiological study.

| Genotype | Animal ID | Age at surgery | Age at experiments ended |
|---|---|---|---|
| CA1-NR1 cKO (*CaMK2-Cre; NR1-flox/flox*) | M18 | 2 months | 3 months |
| | M28 | 3 months | 3 months |
| | M30 | 3 months | 4 months |
| Control (*NR1-flox/flox*) | M24 | 5 months | 5 months |
| | M26 | 3 months | 4 months |
| | M29 | 3 months | 4 months |
| | M31 | 4 month | 5 months |

define the statistical assessment (delay-act cells — significant higher firing rate [rank <50, top 5%]; delay-sup cells — significant lower firing rate [rank >950, bottom 5%]).

## Decoding of delay length from population spike activity

To quantify the information of delay length reflected in the population spike activity, we performed decoding analysis. We used the fitcecoc.m function from MATLAB statistics and the machine-learning toolbox, which enables to train a multiclass, error-correcting output codes (ECOC) model of linear support vector machines (SVM) for binary choices (e.g., *Reber et al., 2019*; *Stavisky et al., 2019*). In this, multiple binary SVMs between all pairs of labels are trained. All parameters were set to their default values. We constructed a feature vector for one or two trials, consisting of the firing activity of each neuron (normalized firing rate [0 to 1]) in 25 bins of 200 ms (over 5 s). The classifier was trained on spike trains from −25 s to 60 s after delay-onset of all five conditions, with labels of the delay length (delay lengths are 0, 5, 10, 20 and 40 s) for each animal (no fewer than 17, not more than 43 neurons from one animal) at every 2 s time step in each trial (*Figure 5—figure supplement 1A*). Classification performance was cross-validated using a leave-one-trial-out method and quantified as the correction probability. We separately calculate the correction probability of each delay length. The performance was shown together with surrogate decoding performance as chance prediction, obtained from artificial testing datasets created by shuffling the neuron labels and/or delay lengths (*Figure 5—figure supplement 1B and D*).

## Statistical analysis

Correlation coefficients and *P* values between firing rates and delay length were calculated by the Matlab function (corrcoef). To estimate statistical significance of the obtained percentage of neurons correlated with delay length, we resampled firing rate and delay length in all trials with 1000 repeats. We then compared the observed percentage from the permutated percentage. To compare the firing rates between short and long delay conditions, we performed Wilcoxon's rank sum test. Kolmogorov–Smirnov test (kstest) (*Salz et al., 2016*) was conducted to test the normality. To assess side-dependency in firing rates, three-way ANOVA (side [right and left] × phase [start, delay, and goal] × delay length [5 and 20]) was used. To compare the effect of reward loss and gain on firing rate of delay-act and delay-sup cells and average firing rates between cKO and control mice, Mann-Whitney's *U* test was carried out. To examine the ratio distribution, we performed two-tailed one-sample *t* tests against 0. The behavioral impact of NMDAR conditional knockout was evaluated by two-way ANOVA (genotype [cKO and control] × choice probability) followed by post-hoc Scheffe's test. Fisher's exact test was applied to compare the cell-type distributions between cKO and control mice.

## Acknowledgements

We thank Drs Steven Middleton, Roman Boehringer, and Chinnakkaruppan Adaikkan for help building tetrodes, Dr Charles Yokoyama for valuable comments, and Dr Susumu Tonegawa for providing us NR1 cKO mice. Funding was provided by a Grand-in-Aid for Exploratory Research (JSPS

KAKENHI Grant Number 16K15196) and from the 'Brain/MINDS' program from the Japan Agency for Medical Research and Development (AMED).

## Additional information

### Funding

| Funder | Grant reference number | Author |
|---|---|---|
| Japan Society for the Promotion of Science | 16K15196 | Akira Masuda |
| Japan Agency for Medical Research and Development | Brain/MINDS | Shigeyoshi Fujisawa |

The funders had no role in study design, data collection and interpretation, or the decision to submit the work for publication.

### Author contributions

Akira Masuda, Conceptualization, Software, Formal analysis, Funding acquisition, Investigation, Visualization, Methodology; Chie Sano, Investigation; Qi Zhang, Validation, Methodology; Hiromichi Goto, Resources, Validation, Methodology; Thomas J McHugh, Conceptualization, Resources, Supervision, Methodology; Shigeyoshi Fujisawa, Conceptualization, Resources, Software, Supervision, Funding acquisition; Shigeyoshi Itohara, Conceptualization, Resources, Data curation, Supervision, Funding acquisition, Validation, Methodology

### Author ORCIDs

Akira Masuda (iD) https://orcid.org/0000-0002-8659-6356

### Ethics

Animal experimentation: This study was performed in strict accordance with the recommendations in the Guide for the Care and Use of Laboratory Animals of the National Institute of Health. The study was approved by the Institutional Animal Care and Use Committee of the RIKEN Institute in Wako (approval number H27-2-239(6)), in conformity with Article 24 of the RIKEN regulations for animal experiments. All surgery was performed under isoflurane anesthesia, and every effort was made to minimize suffering.

### Decision letter and Author response

Decision letter https://doi.org/10.7554/eLife.52466.sa1
Author response https://doi.org/10.7554/eLife.52466.sa2

## Additional files

### Supplementary files

• Transparent reporting form

### Data availability

All data generated or analysed during this study are included in the manuscript and supporting files.

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
