## [Decision Letter]

**Acceptance summary:**

This paper suggests a potential explanation for the known contributions of the hippocampus to adaptive decision-making in the face of temporal delays. Masuda et al. dissociate delay and reward value behaviorally, and are therefore able to show that these decision variables are encoded in distinct populations of neurons in the mouse hippocampus.

**Decision letter after peer review:**

[Editors’ note: the authors submitted for reconsideration following the decision after peer review. What follows is the decision letter after the first round of review.]

Thank you for submitting your work entitled "The hippocampus encodes delay and value information during delay-discounting decision making" for consideration by *eLife*. Your article has been reviewed by three peer reviewers, one of whom is a member of our Board of Reviewing Editors, and the evaluation has been overseen by a Senior Editor. The reviewers have opted to remain anonymous.

Our decision has been reached after consultation between the reviewers. Based on these discussions and the individual reviews below, we regret to inform you that your work will not be considered further for publication in *eLife*.

Although there was broad agreement that the issue of how delay length is encoded in the activity of hippocampal neurons is timely and important, the manuscript lacks the critical analyses needed to evaluate whether there is a meaningful advance towards this goal.

In particular, all reviewers agreed that a permutation test is required to support the important central claim that some number of neurons encode delay length. However, even in the best case scenario where the currently reported number passes such a test, the small number of neurons encoding delay length would limit the significance of the finding. Relatedly, several reviewers felt the analyses performed did not appropriately take the known properties of "time cells" into account. Please see the reviews below for a more detailed explanation of these concerns.

Reviewer #1:

This study by Masuda and colleagues seeks to characterize differences in hippocampal activity during the delay phase of an intertemporal choice task. The question of under what circumstances and through what mechanism the hippocampus contributes to such delay discounting has received considerable attention in the human decision-making literature and in rodent behavioral neuroscience studies. Surprisingly, no studies appear to have asked how the rodent HC represents such choices in ensemble neural activity. Thus, I am excited about the potential of the study to fill a conspicuous neural representation gap in our working model of how the HC contributes to intertemporal choice.

The authors correctly identified that temporal discounting requires encoding of multiple different delay lengths, rather than simply the presence or absence of a delay. They therefore include different delay conditions in their task, so that firing rate and delay length can be correlated in each individual neuron. The main finding appears to be that a small set of neurons (<10% overall; 3 out of 58 and 8 out of 83 for delay-activated and -suppressed neurons respectively) encodes delay length thus defined. I do not find this result very informative for a number of reasons.

First, the authors should perform permutation tests to establish a chance distribution to compare these numbers to.

Second, the data suggest this analysis does not capture the properties of CA1 ensemble activity adequately: the population averages (Figure 4B) show a non-monotonic relationship with delay length, and the peri-event time histograms show the characteristic distribution of peak firing rates across the delay previously shown by Pastalkova, MacDonald, and others. The "retiming" experiments suggest that the sequence of firing reliably distinguishes the different delays - wouldn't this provide a very different affirmative answer to the authors' question of how HC activity encodes delay?

Third, although the authors manipulate reward amount and report how such changes affect delay-activated vs. -suppressed neurons, I could not find how the delay-encoding neurons specifically (the 3/58 and 8/83) were affected by reward amount. This is crucial to establish, because of the possibility that what looks like delay coding could actually be more general expected value coding.

Thus, to convincingly address the seemingly simple question of how HC activity relates to delay length the authors need to confront two tricky issues which are currently not adequately addressed:

1) Considering different coding schemes, such as sustained parametric coding of delay in individual neurons (e.g. short: 1Hz, medium, 2Hz, long: 3Hz), and dynamic sequence patterns that may differentiate between delays (e.g. short: neuron A->B->C, medium: B->D->E, long: C->D->F), and others.

2) Disentangling delay length from known covariates, such as expected value and movement.

Note that I have not commented on the substantial number of other analyses reported, such as activity dependence on location, the temporal distribution of firing rates, and temporal shifts with experience. These are helpful for reference but do not seem to speak to the main question of how different delay lengths are coded. Similarly, a potentially valuable component of the study is the altered behavior and neural activity in CA1 NMDAR knockout mice, but given the issues above, it is presently unclear how this should be interpreted.

Reviewer #2:

In this manuscript, Masuda and colleagues investigate an interesting and important question: how does the hippocampus represent a delay-based decision making task? The authors identify populations of neurons that fired (or ceased to fire) while mice waited in the delay zone, and showed that in a subset of these neurons delay-zone firing rate correlated with delay length. Interestingly, populations of delay-activated and delay-suppressed neurons showed opposite effects when reward size was decreased and then restored. Finally, the authors repeated many of these analyses in NMDA receptor knock-out mice, and found altered patterns of decision making and neural representations in these animals.

Given the wealth of evidence that hippocampus is important for decisions involving delays, an attempt to understand what is going on at the single-unit level during the delay period is an interesting and potentially valuable contribution to the literature. However, the most interesting and novel effects reported here are identified in very small numbers of neurons, raising potential questions about the robustness of the effects. In addition, the logic of the authors' method of identifying delay suppressed and activated neurons is unclear.

Samples sizes and neural effect sizes

Many of the effects reported in the manuscript are identified in small subgroups of neurons. I think this is partially because the authors ran several variants of the task and did not necessarily record large numbers of neurons for each of those different conditions, but even for the main task the authors report that mice typically performed around 10 trials per session, which is not so many for neural data analyses. This means that interesting effects like the potential correlation between firing rate and delay length are identified in rather small populations of cells (3/58 delay activated and 8/83 delay suppressed cells). These number are low, both in terms of the fraction of neurons that show the effect (which is pretty near chance for this example; 5% and 9% of neurons), and the total number of neurons analyzed for each effect, which raises the possibility that the analyses are somewhat underpowered and potentially spurious. Similarly, the delay-dependent shift in firing location (reported in Figure 5) is identified in 4/33 cells. The percentage of neurons whose firing rate changed during the revaluation procedure (where the delayed reward was decreased and then restored back to the normal size) was not reported, but the entire analysis was conducted on 30 delay activated neurons and 39 delay suppressed neurons, which is again a small number of cells to reliably detect changes.

Besides the issue of the number of cells recorded under different conditions, there are other aspects of the data that I don't understand and cannot parse by reading the paper. For instance, in the reward revaluation analysis there are pretty dramatically different numbers of neurons in the devalued and revalued conditions, meaning either that for some reason cells were lost between trials (indicating pretty serious recording instability), or this analysis was actually conducted across different recording sessions (rendering a comparison of absolute different in firing rate fraught, as it's unclear what fraction of neurons were recorded in both sessions). The fact that I'm still not entirely sure how this part of the experiment was carried out points to some issues with the methods description; with some many different variants of the task, it would be nice if more detailed descriptions of each were provided, along with a detailed timeline of which order the variations occurred in.

Approach for detection delay-activated and delay-suppressed cells

Given the naming of these cells, my initial impression was that they were populations of neurons that either increased or decreased their firing rate with increasing or decreasing delay. In fact, only very few neurons in their sample showed that sort of behavior. Instead, these groups of neurons are identified based on comparing their average firing rate in the delay zone with their average firing rate everywhere else on the track. To me, this makes it hard to specifically say the neurons identified in this way were specifically modulated by delay (as their naming implies), because the delay zone has other properties that set it apart from the rest of the track. Presumably animal's movement speed is low here, while it is high everywhere else on the track expect perhaps the reward zone. It's also the part of the maze where mice presumably spend the greatest total amount of session time. To anthropomorphize a bit, it's probably the region of the maze associated with the most frustration or annoyance due to the delay. All of these factor could be reasons for detecting a difference in firing rate in this particular location relative to every other location on the maze. Again, if the change in firing rate were linked directly to systematic variations in delay, I think that would go some ways towards ruling out other possibilities like the ones I mentioned here, but in fact, the data indicate that neurons firing rate correlated with delay length are quite rare.

Reviewer #3:

The authors of the study investigated the activity of CA1 neurons in the mouse during a delay discounting task. The main finding is that the activity of select CA1 neurons can reflect reward temporal delay, amount, and location by delay-activation or delay-suppression of spiking activity in a NMDA-dependent manner. The authors conclude that distinct subclasses of hippocampal neurons support delay-discounting decisions of the animals. The results of the study are novel and interesting and can only suggests several ways to further improve the significance of the current results.

1) Some of the reported effects are rather small (one example: proportion of neurons showing correlation between changes in firing rates and amount of delay at 5.1%, paragraph two of subsection “Delay-dependent neuronal activity in the CA1”). To test the significance of such effects above chance variability, whenever possible, the authors should compare these proportions with those obtained by shuffling neuronal activity, for instance shuffling neuronal identity across different delays, and show the control proportions are lower.

2). There two classes of responsive neurons, delay-act (+) and delay-sup (-). Aside from this feature, is there any other property of these cells that would allow them to be distinguished as two classes of neurons (see Discussion paragraph one)? Related to this, the significant correlations between delay duration and amount of change in firing rate the authors like to emphasize on appear positive (and small) for both delay+ and delay- neurons. Intuitively, I would have expected the interesting correlations to be negative for the delay- neurons and positive, and in higher proportions, for the delay+ neurons. The authors should discuss the significance, importance and implications of these findings.

3) The NR1KO CA1 neurons are known to generally fire with reduced rates compared with control animals. The reported z-scores might become noisier in the mutant animals due to their reduced baseline rates, which could result in reduced proportions of delay+ and delay- neurons compared with controls. The authors should compare the delay+ and delay- activity of NR1KO neurons with that of a subpopulation of control neurons with mean firing rates similar to those of KO ones, in addition to all control neurons.

4) The authors report the recording of inhibitory neurons activity (INT). I suggest the authors further explore this activity in the context of delay+ and delay- activity of putative pyramidal neurons (PYR) as a possible clue to the diversity of neuronal response to delay. For instance, is there any putative synaptic PYR-INT connection detectable in cross-correlations between individuals of these neuronal groups that changes with temporal delay or that could explain the two proposed classes of PYR neurons as well as the effects of NR1 KO?

5) Please indicate which animals were recorded with silicon probes and what kind of probes were used (recording sites configuration).

[Editors’ note: further revisions were suggested prior to acceptance, as described below.]

Thank you for submitting your article "The hippocampus encodes delay and value information during delay-discounting decision making" for consideration by *eLife*. Your article has been reviewed by three peer reviewers, one of whom is a member of our Board of Reviewing Editors, and the evaluation has been overseen by Laura Colgin as the Senior Editor. The reviewers have opted to remain anonymous.

The reviewers have discussed the reviews with one another and the Reviewing Editor has drafted this decision to help you prepare a revised submission.

Summary:

The reviewers agree that the main conclusion of the paper, that there are distinct signals related to delay and value in the activity of CA1 neurons, is now supported by the improved analyses in this revision. They also highlighted two remaining issues that the authors should be able to address with existing data and careful attention to communicating the logic and motivation for the analyses in the text.

Essential revisions:

First, the authors should add an up-front discussion of the possible ways in which delays could be coded in principle, followed by justification of the specific analysis methods chosen. It should be clear to the reader how the methods used are able to identify which of these schemes are supported by the data. See reviewer #1 for more details.

Second, more information and discussion related to the possibility that the NMDA receptor knockout may have spread beyond the hippocampus proper should be added (see reviewer #2).

Reviewer #1:

In this resubmission, the authors have added a number of important additional analyses, in particular (1) permutation tests to establish the statistical significance of the main results, and (2) a direct comparison of delay coding and "reward coding", showing that putative delay coding is not a consequence of temporal discounting of reward (Figure 7E).

A further major change is that the cell numbers (and percentages) coding delay is now an order of magnitude higher than in the original submission, a difference the authors explain is due to computing correlations between delay and firing rate on a trial-by-trial basis, rather than by first averaging across trials.

With these additions, I think the authors have provided solid evidence that the firing rates of a substantial proportion of CA1 neurons is parametrically related to delay length in the main "extension" version of the task.

However, as I wrote in my original review, the authors still do not seem to give consideration to the multitude of ways CA1 population activity can be said to encode delay length. There are previous findings in the literature (such as the MacDonald et al. and follow-up "time cell" papers) that lay out a specific "sequential activation" scheme for what delay activity could look like. It may be that this is not what happens in this data set, but they need to explicitly identify this possibility and then treat it with corresponding analyses.

More generally, the paper really needs an up-front discussion of the possible ways, neural coding schemes, in which delays could be coded in principle, followed by motivation/justification of the specific analysis methods that are able to identify which of these schemes are compatible with the data. Without this, many of the analyses lack a clear logical connection to possible interpretations. Just as one example, suppose that delay activity were to look as follows: short delay is coded by sequential activation of cells A-B-C, longer delay by A-B-C-D-E, and longest delay by A-B-C-D-E-F-G. The “mean firing rate” of cell A will now be parametrically related to delay length, because the same spiking activity will be normalized by a different length time window. I'd say this would be a misleading way of claiming that cell A encodes delay length, because activity isn't actually different between the three delays! I am aware the authors also use peak firing rate, which avoids this particular pitfall (but has other issues, how would a downstream decoder read this out?), but I'm using this example to hopefully impress on the authors the need to clearly motivate their choice of analysis.

There are other points in the manuscript where I found the logic difficult to follow. For instance, I think the analyses in Figure 7E is the logical next step after having shown that there is (in a certain number of cells) a relationship between delay and firing rate. Given that initial result, I would want to know to what extent such a relationship could be the result of correlated/confounding variables such as discounted reward value. In other words, are these neurons "just" coding value? From the authors' rebuttal, I get the impression that I somehow did not make this point fully clear. I will try again. Even though the actual outcome, number of reward pellets, is not changed, a change in delay means that for longer delays, the subjective discounted value is smaller. Thus, in the basic "extension" design, delay length is perfectly (inversely) correlated with subjective value, and it is therefore essential that the authors test if such a value-based account is the best explanation. It is a strength of the study that the authors have data that can address this, but the importance and logic of this argument currently is not clear from the paper, and the fact that the key result is "buried" in Figure 7E does the paper a disservice.

Reviewer #2:

In this revised version of the manuscript, the authors have addressed satisfactorily all my comments. The main concern regarding the low number of coding neurons has been addressed with new data analysis and the proportions of significant neurons are more convincing. Moreover, key terminology has been revised according to reviewers' comments and additional explanations have been added as requested. Overall, the manuscript has improved significantly. There remain several typos throughout the manuscript, mostly on the newly added text. The authors should proofread the manuscript and fix all these errors. Given the completion of that process, I have no further comments and I recommend the manuscript for publication.

I would like for the authors to include a new Table showing the ages of KO animals whose electrophysiological activity was reported in the manuscript (Figure 8, Figure 8—figure supplement 1). This is important as the authors state that in some of these animals the deletion of NMDAR spread beyond the CA1 area. The authors should also perform and report data analyses that are restricted to KO animals that were between 1-2 months of age at the time of ephys recording when NMDA deletion is restricted to CA1 area. If the authors want to maintain the statement currently in the Abstract and throughout the manuscript that "genetic deletion of NMDA receptor in hippocampal pyramidal cells impaired delay-discount behavior and diminished delay-dependent activity in CA1", they should show that this is indeed the case in the subgroup of animals where the NMDA KO was restricted to CA1 pyramidal neurons or at least the hippocampus (not entorhinal cortex or other brain areas). Otherwise, the authors should disclose that the reported effects might be contributed by NMDA deletion outside the hippocampus and name those brain areas. It would also really help if the demonstration of CA1 specific deletion of NMDAR in KO animals used in the ephys could be supported by immunohistochemistry.

---

## [Author Response]

[Editors’ note: the authors resubmitted a revised version of the paper for consideration. What follows is the authors’ response to the first round of review.]

Reviewer #1:This study by Masuda and colleagues seeks to characterize differences in hippocampal activity during the delay phase of an intertemporal choice task. The question of under what circumstances and through what mechanism the hippocampus contributes to such delay discounting has received considerable attention in the human decision-making literature and in rodent behavioral neuroscience studies. Surprisingly, no studies appear to have asked how the rodent HC represents such choices in ensemble neural activity. Thus, I am excited about the potential of the study to fill a conspicuous neural representation gap in our working model of how the HC contributes to intertemporal choice.The authors correctly identified that temporal discounting requires encoding of multiple different delay lengths, rather than simply the presence or absence of a delay. They therefore include different delay conditions in their task, so that firing rate and delay length can be correlated in each individual neuron. The main finding appears to be that a small set of neurons (<10% overall; 3 out of 58 and 8 out of 83 for delay-activated and -suppressed neurons respectively) encodes delay length thus defined. I do not find this result very informative for a number of reasons.First, the authors should perform permutation tests to establish a chance distribution to compare these numbers to.Second, the data suggest this analysis does not capture the properties of CA1 ensemble activity adequately: the population averages (Figure 4B) show a non-monotonic relationship with delay length, and the peri-event time histograms show the characteristic distribution of peak firing rates across the delay previously shown by Pastalkova, MacDonald, and others. The "retiming" experiments suggest that the sequence of firing reliably distinguishes the different delays – wouldn't this provide a very different affirmative answer to the authors' question of how HC activity encodes delay?Third, although the authors manipulate reward amount and report how such changes affect delay-activated vs. -suppressed neurons, I could not find how the delay-encoding neurons specifically (the 3/58 and 8/83) were affected by reward amount. This is crucial to establish, because of the possibility that what looks like delay coding could actually be more general expected value coding.

a) We thank the reviewer for these excellent suggestions, they have motivated us to rethink our analytical approach to identifying and verifying delay activity in CA1. First, in response to comments raised by all three reviewers, we have updated our approach to detect neurons showing correlation with delay length. In our original manuscript, the correlation analysis only used the averaged firing rate for all trials of each of the four delay lengths, reducing each neuron’s activity to four data points and making it very difficult to identify a meaningful number of single units (only 4 variables, R = 0.96 was required to reach significance P < 0.05). To make use of the rich data set we collected, we reanalyze correlation coefficients using each trial as a sample, and examined the delay length/firing rate changes using both the mean and peak firing rate of each cell separately. In this careful analysis we found a substantial fraction of the neurons demonstrated a significant correlation with delay length (about 70% for peak firing rate; about 35% for mean firing rate, Figure 5 in the revised manuscript). As you suggested, we performed a permutation test to simulate the chance probability of cell showing correlation and confirmed that the chance level (10.8 – 12.7% ) is much lower than what we observed.

b) As for the second point you raise, we agree that the population firing rate (Figure 4B in original manuscript) does in fact show a non-monotonic response across the longer delay, which highlights different trends among the delay-act and -sup cells. As we argued above, here we identify a novel group of neurons which increased or decreased their firing rate proportionally to the delay length. Thus, this finding suggests that not all delay-act or -sup cells, but rather a certain subset is involved in encoding of delay length.

In the terms of sequence structure observed in the population analysis, we conducted an additional analysis. We separately generated population codes of different experimental conditions. For estimation of factors in population structure, we separately produced peri-event time histograms in the different experimental conditions (extension, switch, both-side, and reward lose conditions). Interestingly we found a uniform-like distribution only from both-side conditions (Kolmogorov–Smirnov test, kstest; Salz et al., 2016, did not deny the possibility of uniformity in both-side conditions but did other all conditions, p = 0.46 for both-side condition). This suggests that CA1 population activity is diverse across the experimental conditions. Furthermore, uniform-like distribution of CA1 neurons may require repeated presentation of a constant delay. We add this information in Figure 4—figure supplement 2 and text.

c) From our recording data, CA1 cells could be analytically categorized with i) activity during delay (delay act and -sup), ii) responsiveness to long delay (up or down, and correlation), iii) place-dependency (side-dependent or -independent), and iv) reward dependency (up or down by reward loss or gain). For iii) and iv), we designed the experiment scheme to make it possible to cross-evaluate these responses with i) and ii). In reward loss or gain conditions (in original manuscript, devalue or revalue conditions), the CA1 activity was recorded under short delay and long delay, as well as under the reward manipulated conditions (see Figure 7—figure supplement 1 in the revised version). Thus, we could analyze the relationship among the delay-related activity and reward manipulation (Figure 7E in the revised version). The scatter plot of firing rate ratio small/large reward conditions and long/short delay conditions shows that there are no global trends (note that delay-act cells tend to be negative on small/large axis), suggesting that the CA1 neurons exhibit independent relationship delay (length) and reward manipulation responses.

Thus, to convincingly address the seemingly simple question of how HC activity relates to delay length the authors need to confront two tricky issues which are currently not adequately addressed:1) Considering different coding schemes, such as sustained parametric coding of delay in individual neurons (e.g. short: 1Hz, medium, 2Hz, long: 3Hz), and dynamic sequence patterns that may differentiate between delays (e.g. short: neuron A->B->C, medium: B->D->E, long: C->D->F), and others.2) Disentangling delay length from known covariates, such as expected value and movement.

Thank you for your critical perspective. As per the reviewer’s suggestion, we have reconceptualized our idea of delay coding in CA1. You have raised the idea of coding schemes based on the individual and population levels. Our data showed that a large fraction of neurons in CA1 reflect delay length in their firing rate. This would be the individual level of delay coding. At the population level, the uniform-like distribution of peak firing over task time was observed under a specific protocol, the both-side condition, where a constant delay length repeatedly presented. This suggests that the CA1 activity is controlled in a sequential manner to encode precise time. We added this point of view to the Discussion.

**Author response image 1. respfig1:** Moving patterns and speed during delay. Left: Moving traces of single trial on delay 5, 10, 20 and 40 s conditions. Right: Moving speed corresponding to the trials shown in the left traces. Initial high speed movements (around 20-24 cm/s) occurred by starting a trial are followed by constant intermitted speed movements (around 16 cm/s). Note that there is no obvious difference in moving speed among the delay 5, 10, 20 and 40 s conditions. Data from mouse M9.

Thank you for the suggestion. Covariates such as reward expectation and spatial context were investigated in an integrative manner. We believe understanding “compositeness” for CA1 activity is important because the CA1 activity has already known to be related to single components such as movement speed, reward expectation, and spatial context. The expected values in the present study were constant because the percentage of reward is always 100% within the all trials, although some initial trials after transition of switch, both-side, or reward loss/gain conditions perturbed the animals’ prediction of reward size. To consider this issue, we have shown the firing in all trials for such conditions and followed the transition and adaptation in a qualitative manner. Regarding moving speed, we captured example data in the bottom. It was general that the animals were running inside the delay zone during waiting. The movement trajectories and speed shown here do not match the firing patterns of CA1 neurons. Strong theta power during that time (Figure 2C-D) also suggest that the animals waiting in the delay zone are active in brain state.

Note that I have not commented on the substantial number of other analyses reported, such as activity dependence on location, the temporal distribution of firing rates, and temporal shifts with experience. These are helpful for reference but do not seem to speak to the main question of how different delay lengths are coded. Similarly, a potentially valuable component of the study is the altered behavior and neural activity in CA1 NMDAR knockout mice, but given the issues above, it is presently unclear how this should be interpreted.

We tried to explain importance of all analysis in the revised manuscript.

Reviewer #2:In this manuscript, Masuda and colleagues investigate an interesting and important question: how does the hippocampus represent a delay-based decision making task? The authors identify populations of neurons that fired (or ceased to fire) while mice waited in the delay zone, and showed that in a subset of these neurons delay-zone firing rate correlated with delay length. Interestingly, populations of delay-activated and delay-suppressed neurons showed opposite effects when reward size was decreased and then restored. Finally, the authors repeated many of these analyses in NMDA receptor knock-out mice, and found altered patterns of decision making and neural representations in these animals.Given the wealth of evidence that hippocampus is important for decisions involving delays, an attempt to understand what is going on at the single-unit level during the delay period is an interesting and potentially valuable contribution to the literature. However, the most interesting and novel effects reported here are identified in very small numbers of neurons, raising potential questions about the robustness of the effects. In addition, the logic of the authors' method of identifying delay suppressed and activated neurons is unclear.

The reviewer raises a key point, also referenced by the other reviewers. The small population of neurons the reviewers latch onto (3 out of 58 and 8 out of 83 for delay-activated and -suppressed neurons respectively) were CA1 neurons which showed a statistically significant correlation between delay length and average firing rate change across all tested delay conditions (5, 10, 20, and 40 sec). This number was small because we used only average firing rate over whole trials from each delay length condition, making it very difficult to obtain statistically significant cells when sample size is only four points (firing rate at delay length of 5, 10, 20, and 40 sec) for correlation analysis (R = 0.96 is required). Therefore, we have approached this question with a new analysis, now using the richness of our data set to include firing rates individual from all delay trials (about 30 – 40 trials). In this correlation analysis, we found a significant fraction of neurons which showed significant correlation (P < 0.05) of peak firing rate (about 70% ) and mean firing rate (about 35% ) with delay length (Figure 5). A permutation test, randomization of firing and delay length pairs, confirmed the percentages are higher than chance level (10.8 – 12.7% , P = 0.01).

As an answer to the second question, the confusion may stem from our admittedly poor overall definition of delay-act and delay-sup cells. We categorized them based on whether “the firing rate during the long delays (>20 sec)” is higher than the “baseline” (activity during delay is high or low) with applying a permutation test (compared to randomized firing). We apologize that some reviewers were confused by our finding that the delay-act cells from the neurons of firing rate significantly facilitated by delay length extension (short to long) and the delay-sup cells suppressed. This characteristic (facilitation or inhibition by delay extension) is investigated by comparison short and long delay conditions throughout our study. Therefore, we followed the firing pattern of CA1 neurons two directions: within trial (delay-act and delay-sup cells, selective activity to “delay”) and among different delay lengths (short: 5 sec and long: 20 sec, or 0, 5, 10, 20, and 40 sec).

Samples sizes and neural effect sizesMany of the effects reported in the manuscript are identified in small subgroups of neurons. I think this is partially because the authors ran several variants of the task and did not necessarily record large numbers of neurons for each of those different conditions, but even for the main task the authors report that mice typically performed around 10 trials per session, which is not so many for neural data analyses. This means that interesting effects like the potential correlation between firing rate and delay length are identified in rather small populations of cells (3/58 delay activated and 8/83 delay suppressed cells). These number are low, both in terms of the fraction of neurons that show the effect (which is pretty near chance for this example; 5% and 9% of neurons), and the total number of neurons analyzed for each effect, which raises the possibility that the analyses are somewhat underpowered and potentially spurious. Similarly, the delay-dependent shift in firing location (reported in Figure 5) is identified in 4/33 cells. The percentage of neurons whose firing rate changed during the revaluation procedure (where the delayed reward was decreased and then restored back to the normal size) was not reported, but the entire analysis was conducted on 30 delay activated neurons and 39 delay suppressed neurons, which is again a small number of cells to reliably detect changes.

We apologize for the confusion. The number of analyzed neurons in these experiments were more than 800 in total. We think this number is at the high end for electrophysiological recordings in free moving mice. The reviewer is correct in that the number of neurons recorded under the various conditions was variable, we have included Table 2 that clarifies the number of mice and the number of neurons recorded under each individual condition. Further, as mentioned above, the small number of neurons in some of the analyses was due to several reasons. First, we divided whole population into delay-act and -sup cells for profiling subcategorized population. Second, neurons were recorded in the different experimental conditions which all include short and long delay conditions, although more than 80 neurons identified in each condition. Third, we only included neurons with >0.5 Hz average firing rate because some statistical methods, including permutation test for definition of delay-act and -sup neurons, require relatively higher firing rates. We believe that despite these challenges, the numbers are valid to assess statistical significance. Regarding the issue that small fraction of significantly correlated with delay length in firing rate, please see our answer to point 1 above.

Besides the issue of the number of cells recorded under different conditions, there are other aspects of the data that I don't understand and cannot parse by reading the paper. For instance, in the reward revaluation analysis there are pretty dramatically different numbers of neurons in the devalued and revalued conditions, meaning either that for some reason cells were lost between trials (indicating pretty serious recording instability), or this analysis was actually conducted across different recording sessions (rendering a comparison of absolute different in firing rate fraught, as it's unclear what fraction of neurons were recorded in both sessions). The fact that I'm still not entirely sure how this part of the experiment was carried out points to some issues with the methods description; with some many different variants of the task, it would be nice if more detailed descriptions of each were provided, along with a detailed timeline of which order the variations occurred in.

The relatively small sample size of neurons in the reward manipulation conditions stemmed from the experimental schedule which is now referenced in Figure 1—figure supplement 2. We sequentially performed different experimental conditions, and the reward manipulation conditions were at the end of the whole schedule. The number of neurons usually dropped day by day. We added the daily schedule of each experiment as figure supplements.

Approach for detection delay-activated and delay-suppressed cellsGiven the naming of these cells, my initial impression was that they were populations of neurons that either increased or decreased their firing rate with increasing or decreasing delay. In fact, only very few neurons in their sample showed that sort of behavior. Instead, these groups of neurons are identified based on comparing their average firing rate in the delay zone with their average firing rate everywhere else on the track. To me, this makes it hard to specifically say the neurons identified in this way were specifically modulated by delay (as their naming implies), because the delay zone has other properties that set it apart from the rest of the track. Presumably animal's movement speed is low here, while it is high everywhere else on the track expect perhaps the reward zone. It's also the part of the maze where mice presumably spend the greatest total amount of session time. To anthropomorphize a bit, it's probably the region of the maze associated with the most frustration or annoyance due to the delay. All of these factor could be reasons for detecting a difference in firing rate in this particular location relative to every other location on the maze. Again, if the change in firing rate were linked directly to systematic variations in delay, I think that would go some ways towards ruling out other possibilities like the ones I mentioned here, but in fact, the data indicate that neurons firing rate correlated with delay length are quite rare.

Thank you for the wide range of suggestions. The delay in the task has two aspects: which option (or position) is associated, and how long the animals need to wait. Although the delay is not always long in the task, delay length did have a strong impact on choice behavior (Figure 1). To define delay-act cells we set our criteria based on the activity during delays longer than 20 s, which had a stronger effect of dropping animals’ preference to the delayed reinforcer. As the reviewer suggests, we could also define delay-act cells based on the enhanced firing rate after delay extensions (such as delay 5 to 20 sec). However, it would be more complex when the activity is compared to the different conditions tested in the present study, such as switch or both-side conditions where delay length changed in multiple position simultaneously. Moreover, we measure the change of firing patterns by delay extensions in some experimental conditions including extension and reward loss and gain conditions. This information is shown in the correlation analyses.

We agree with the reviewer’s opinion that the several factors may induce neural activity changes. We checked one possibility that difference of the moving speed during delay may affect firing activity (see Author response image 1). From this figure showing the examples of moving speed of different delay length, the animals seem to move at constant moving speed over delay 5,10,20, and 40 sec conditions.

Reviewer #3:The authors of the study investigated the activity of CA1 neurons in the mouse during a delay discounting task. The main finding is that the activity of select CA1 neurons can reflect reward temporal delay, amount, and location by delay-activation or delay-suppression of spiking activity in a NMDA-dependent manner. The authors conclude that distinct subclasses of hippocampal neurons support delay-discounting decisions of the animals. The results of the study are novel and interesting and can only suggests several ways to further improve the significance of the current results.1) Some of the reported effects are rather small (one example: proportion of neurons showing correlation between changes in firing rates and amount of delay at 5.1%, paragraph two of subsection “Delay-dependent neuronal activity in the CA1”). To test the significance of such effects above chance variability, whenever possible, the authors should compare these proportions with those obtained by shuffling neuronal activity, for instance shuffling neuronal identity across different delays, and show the control proportions are lower.

The small population of neurons that all the reviewers expressed concern with were CA1 neurons which showed a statistically significant correlation between delay length and mean firing rate change across all tested delay conditions (3 out of 58 and 8 out of 83 for delay-activated and -suppressed neurons respectively). This number was small because we used only average firing rate over whole trials from each delay length condition, making it difficult to obtain statistically significant cells when sample size is only four (firing rate at delay length of 5, 10, 20, and 40 sec) for correlation analysis (R = 0.96 is required). Therefore, we employed the firing rate of each neuron on a trial-by-trial basis over the delay trials (about 30 – 40 trials). In this correlation analysis, we found a large fraction of neurons showed significant correlation of (P < 0.05) peak firing rate (about 70% ) and mean firing rate (about 35% ) with delay length. A permutation test, randomization of firing and delay length pairs, confirmed the percentages are higher than chance level (10.8 – 12.7% , P = 0.01).

2). There two classes of responsive neurons, delay-act (+) and delay-sup (-). Aside from this feature, is there any other property of these cells that would allow them to be distinguished as two classes of neurons (see Discussion paragraph one)? Related to this, the significant correlations between delay duration and amount of change in firing rate the authors like to emphasize on appear positive (and small) for both delay+ and delay- neurons. Intuitively, I would have expected the interesting correlations to be negative for the delay- neurons and positive, and in higher proportions, for the delay+ neurons. The authors should discuss the significance, importance and implications of these findings.

Thank you for the important suggestions. We provide a revised figure (Figure 5) addressing these points. It shows that many neurons exhibit a negative correlation of their mean firing rate and delay length during the delay. Our interpretation of these data is that the longer delay is encoded via two different methods: a discounting factor (or delay length, we think these have similar meanings) and predicting factor for larger rewards. The neurons showing negative correlation may be candidate as later one. We added this discussion in the text.

3) The NR1KO CA1 neurons are known to generally fire with reduced rates compared with control animals. The reported z-scores might become noisier in the mutant animals due to their reduced baseline rates, which could result in reduced proportions of delay+ and delay- neurons compared with controls. The authors should compare the delay+ and delay- activity of NR1KO neurons with that of a subpopulation of control neurons with mean firing rates similar to those of KO ones, in addition to all control neurons.

According to the literature (McHugh et al., 1996), the firing rate of NR1cKO CA1 neurons are increased compared to controls. In our data, mean firing rate of whole population CA1 neurons were not significantly different. Following text was add in the result section: “We could not detect significant difference among the genotypes in basic firing property during the task (mean firing rate, cKO: 3.07 + Hz; Control: 3.39 Hz; Z = -0.76, P = 0.44, Mann-Whitney’s U-test). Subpopulation of firing rate were also not significantly different (delay-act, cKO: 2.66 Hz, Control: 3.13 Hz, Z = -0.91, P = 0.35; delay-sup, cKO: 3.47 Hz, Control: 3.47 Hz, Z = -0.68, P = 0.14, Mann-Whitney’s U-test).”

4) The authors report the recording of inhibitory neurons activity (INT). I suggest the authors further explore this activity in the context of delay+ and delay- activity of putative pyramidal neurons (PYR) as a possible clue to the diversity of neuronal response to delay. For instance, is there any putative synaptic PYR-INT connection detectable in cross-correlations between individuals of these neuronal groups that changes with temporal delay or that could explain the two proposed classes of PYR neurons as well as the effects of NR1 KO?

Thank you for the suggestion. We attached data of inhibitory neurons regarding population coding (Figure 4—figure supplement 1).

5) Please indicate which animals were recorded with silicon probes and what kind of probes were used (recording sites configuration).

We understand the importance of correspondence among animal identity, recording sites, and probe design. We added such information in Figure 2—figure supplement 1.

[Editors’ note: what follows is the authors’ response to the second round of review.]

Essential revisions:First, the authors should add an up-front discussion of the possible ways in which delays could be coded in principle, followed by justification of the specific analysis methods chosen. It should be clear to the reader how the methods used are able to identify which of these schemes are supported by the data. See reviewer #1 for more details.

We appreciate this suggestion. First, we have added a leading discussion about the framework of hypothetical coding schemes (population codes and rate coding in individual neurons for delay coding) in the Introduction. Next, we added an analysis to ask if the delay lengths can be decoded by spike activity in population of CA1 neurons (Figure 5—figure supplement 1). The results showed that a machine learning method (multiclass classification learning using SVM) successfully decoded delay length by population spikes at a level much higher than chance. Thus, we could provide evidence that the population activity in the hippocampus may encode delay length. In the main structure of the manuscript, we have strengthened the support for the conclusion that individual neural codes for delay length are modulated by information such as reward and spatial relationship. This clarifies that Figure 7E, demonstrating the feature of individual neurons associated with delay increment and reward amounts reduction, could be interpreted as value loss, a common concept between these manipulations. This allows us to demonstrate that there were a proportion of neurons which meet a requirement of value coding.

Second, more information and discussion related to the possibility that the NMDA receptor knockout may have spread beyond the hippocampus proper should be added (see reviewer #2).

We are happy to include further information of the NMDA receptor knockout and control mice used for electrophysiology. We added a table (Table 5) indicating their ages at surgery and at the end of experiments and added further discussion about this issue in the text. It is important to note that all behavioral experiments were conducted with 2 months old mice, an age at which the KO is specific to CA1 (Fukaya et al., 2003).

Reviewer #1:In this resubmission, the authors have added a number of important additional analyses, in particular (1) permutation tests to establish the statistical significance of the main results, and (2) a direct comparison of delay coding and "reward coding", showing that putative delay coding is not a consequence of temporal discounting of reward (Figure 7E).A further major change is that the cell numbers (and percentages) coding delay is now an order of magnitude higher than in the original submission, a difference the authors explain is due to computing correlations between delay and firing rate on a trial-by-trial basis, rather than by first averaging across trials.With these additions, I think the authors have provided solid evidence that the firing rates of a substantial proportion of CA1 neurons is parametrically related to delay length in the main "extension" version of the task.However, as I wrote in my original review, the authors still do not seem to give consideration to the multitude of ways CA1 population activity can be said to encode delay length. There are previous findings in the literature (such as the MacDonald et al. and follow-up "time cell" papers) that lay out a specific "sequential activation" scheme for what delay activity could look like. It may be that this is not what happens in this data set, but they need to explicitly identify this possibility and then treat it with corresponding analyses.More generally, the paper really needs an up-front discussion of the possible ways, neural coding schemes, in which delays could be coded in principle, followed by motivation/justification of the specific analysis methods that are able to identify which of these schemes are compatible with the data. Without this, many of the analyses lack a clear logical connection to possible interpretations. Just as one example, suppose that delay activity were to look as follows: short delay is coded by sequential activation of cells A-B-C, longer delay by A-B-C-D-E, and longest delay by A-B-C-D-E-F-G. The “mean firing rate” of cell A will now be parametrically related to delay length, because the same spiking activity will be normalized by a different length time window. I'd say this would be a misleading way of claiming that cell A encodes delay length, because activity isn't actually different between the three delays! I am aware the authors also use peak firing rate, which avoids this particular pitfall (but has other issues, how would a downstream decoder read this out?), but I'm using this example to hopefully impress on the authors the need to clearly motivate their choice of analysis.

We added a leading discussion about the framework of hypothetical coding schemes, population codes (sequential activation of multiple neurons/time cells) and rate coding in individual neurons to the Introduction.

An analysis we performed in the revised manuscript asked whether the delay lengths can be decoded by spiking activity across the population. We found that a machine learning method using support vector machine (SVM) successfully decoded different conditions of delay lengths (0, 5, 10, 20, and 40 s) by population spikes (50-60% accuracy, at 20% chance level). Thus, we could provide evidence that the population activity in the hippocampus may encode delay length. This result is shown as Figure 5—figure supplement 1.

In the main structure of the manuscript, we have strengthened the possibility of individual neural codes for delay length in interacting with other information such as reward and spatial relationship, extending the interpretation of Figure 7E, the feature of individual neurons associated with delay codes and reward amounts.

There are other points in the manuscript where I found the logic difficult to follow. For instance, I think the analyses in Figure 7E is the logical next step after having shown that there is (in a certain number of cells) a relationship between delay and firing rate. Given that initial result, I would want to know to what extent such a relationship could be the result of correlated/confounding variables such as discounted reward value. In other words, are these neurons "just" coding value? From the authors' rebuttal, I get the impression that I somehow did not make this point fully clear. I will try again. Even though the actual outcome, number of reward pellets, is not changed, a change in delay means that for longer delays, the subjective discounted value is smaller. Thus, in the basic "extension" design, delay length is perfectly (inversely) correlated with subjective value, and it is therefore essential that the authors test if such a value-based account is the best explanation. It is a strength of the study that the authors have data that can address this, but the importance and logic of this argument currently is not clear from the paper, and the fact that the key result is "buried" in Figure 7E does the paper a disservice.

Thank you for the critical remark. Indeed, there was a decrease in subjective value with correlation to delay length in extension conditions, but it is hard to interpret using only the data of this conditions. One reason is the well-established fact that hippocampal activity is strongly related with the spatiotemporal events. For example, the delay-length correlated neurons we found in the extension condition can be also interpreted as correlated with subjective value (and named as value correlated neurons or so). The subjective value is defined by cost-benefit integration, and delay must be recognized as a factor of cost in the task. Thus, delay and reward are common factors which modulate subjective value. In our opinion, the most important question is if the decrease of value by delay increment and that by reward reduction are similar or not in the rate response of CA1 neurons. If some neurons show reduced “common” firing rate change in response to delay increment and reward loss, the neurons can be recognized as value coding. Although the majority of CA1 neurons did not fit this criterion, there were some CA1 neurons that met this characteristic (plotted around the line of “delay effect = reward effect” in revised Figure 7E). We like to emphasize that these neurons showed common responses to delay increment and reward reduction, thus can be interpreted as encoding subjective value, a “common” concept. We tried to explain this logic more clearly in the Introduction. Result, and Discussion.

Reviewer #2:In this revised version of the manuscript, the authors have addressed satisfactorily all my comments. The main concern regarding the low number of coding neurons has been addressed with new data analysis and the proportions of significant neurons are more convincing. Moreover, key terminology has been revised according to reviewers' comments and additional explanations have been added as requested. Overall, the manuscript has improved significantly. There remain several typos throughout the manuscript, mostly on the newly added text. The authors should proofread the manuscript and fix all these errors. Given the completion of that process, I have no further comments and I recommend the manuscript for publication.I would like for the authors to include a new Table showing the ages of KO animals whose electrophysiological activity was reported in the manuscript (Figure 8, Figure 8—figure supplement 1). This is important as the authors state that in some of these animals the deletion of NMDAR spread beyond the CA1 area. The authors should also perform and report data analyses that are restricted to KO animals that were between 1-2 months of age at the time of ephys recording when NMDA deletion is restricted to CA1 area. If the authors want to maintain the statement currently in the Abstract and throughout the manuscript that "genetic deletion of NMDA receptor in hippocampal pyramidal cells impaired delay-discount behavior and diminished delay-dependent activity in CA1", they should show that this is indeed the case in the subgroup of animals where the NMDA KO was restricted to CA1 pyramidal neurons or at least the hippocampus (not entorhinal cortex or other brain areas). Otherwise, the authors should disclose that the reported effects might be contributed by NMDA deletion outside the hippocampus and name those brain areas. It would also really help if the demonstration of CA1 specific deletion of NMDAR in KO animals used in the ephys could be supported by immunohistochemistry.

We thank the reviewer for further important suggestions. We appended a table (Table 5) showing the ages of cKO and control animals for the electrophysiological study. The ages were 2-4 months old for cKO and 3-5 months old for control mice. We also described the possibility of the spread of the NR1 knockout in the electrophysiological results. It is important to note that all behavioral experiments were conducted with 2 months old mice, an age at which the KO is specific to CA1 (Fukaya et al., 2003). However, as the physiology was conducted up to 4 months of age and we do not wish to overstate the effect on neural activity, we have noted this limitation of the study in the revised Discussion.